



# Integrated airborne investigation of the air composition over the Russian Sector of the Arctic

Boris D. Belan[1], Gerard Ancellet[2], Irina S. Andreeva[3], Pavel N. Antokhin[1], Viktoria G. Arshinova[1], Mikhail Y. Arshinov[1], Yurii S. Balin[1], Vladimir E. Barsuk[4], Sergei B. Belan[1], Dmitry G. Chernov[1], Denis K. Davydov[1], Alexander V. Fofonov[1], Georgii A. Ivlev[1], Sergei N. Kotel'nikov[5], Alexander S. Kozlov[6], Artem V. Kozlov[1], Katharine Law[2], Andrey V. Mikhal'chishin[4], Igor A. Moseikin[4], Sergei V. Nasonov[1], Philippe Nédélec[7], Olesya V. Okhlopkova[3], Sergei E. Ol'kin[3], Mikhail V. Panchenko[1], Jean-Daniel Paris[8], Iogannes E. Penner[1], Igor V. Ptashnik[1], Tatyana M. Rasskazchikova[1], Irina K. Reznikova[3], Oleg A. Romanovskii[1], Alexander S. Safatov[3], Denis E. Savkin[1], Denis V. Simonenkov[1], Tatyana K. Sklyadneva[1], Gennadii N. Tolmachev[1], Semyon V. Yakovlev[1], Polina N. Zenkova[1]

[1]V.E. Zuev Institute of Atmospheric Optics SB RAS, Tomsk, 634055, Russia
[2] Laboratoire Atmosphères, Milieux, Observations Spatiales, LATMOS, UMR 8190, Paris, 78035, France
[3]VECTOR State Research Center of Virology and Biotechnology, Rospotrebnadzor, Koltsovo, Novosibirsk oblast, 630559, Russia
[4]S.A. Chaplygin Siberian Aeronautical Research Institute, Novosibirsk, 630051, Russia
[5]Prokhorov General Physics Institute RAS, Moscow, 119991, Russia
[6]Voevodsky Institute of Chemical Kinetics and Combustion SB RAS, Novosibirsk, 630090, Russia
[7]Laboratoire d'Aerologie, CNRS-UPS, Toulouse, France
[8]Laboratoire des Sciences du Climat et de l'Environnement, LSCE/IPSL, CNRS-CEA-UVSQ, Orme des Merisiers, CEA Saclay, Gif-sur-Yvette, 91191, France

*Correspondence to*: Oleg A. Romanovskii (roa@iao.ru)

**Abstract.** The change of the global climate is most pronounced in the Arctic, where the air temperature increases two to three times faster than the global average. This process is associated with an increase in the concentration of greenhouse gases in the atmosphere. There are publications predicting the sharp increase of methane emissions into the atmosphere due to permafrost thawing. Therefore, it is important to study how the air composition in the Arctic changes in the changing climate. In the Russian sector of the Arctic, the air composition was measured only in the surface atmospheric layer at the coastal stations or earlier at the drifting stations. Vertical distributions of gas constituents of the atmosphere and aerosol were determined only in few small regions. That is why the integrated experiment was carried out to measure the composition of the troposphere in the entire Russian sector of the Arctic from onboard the Optik Tu-134 aircraft laboratory in the period of September 4 to 17 of 2020. The aircraft laboratory was equipped with contact and remote measurement facilities. The contact facilities were capable of measuring the concentrations of $CO_2$, $CH_4$, $O_3$, $CO$, $NO_X$, and $SO_2$, as well as the disperse composition of particles in the size range from 3 nm to 32 μm, black carbon, organic and inorganic components of atmospheric aerosol. The remote facilities were operated to measure the water transparency in the upper layer of the ocean, the chlorophyll content in water, and spectral characteristics of the underlying surface. The measured data have shown that the ocean continues absorbing $CO_2$. This process is most intense over the Barents and Kara Seas. The recorded methane



concentration was increased over all the arctic seas, reaching 2090 ppb in the near-water layer over the Kara Sea. The contents of other gas components and black carbon were close to the background level.

In bioaerosol, bacteria predominated among the identified microorganisms. In most samples, they were represented by coccal forms, less often spore-forming and non-spore-bearing rod-shaped bacteria. No dependence of the representation of 40  various bacterial genera on the height and the sampling site was revealed. The most turbid during the experiment was the upper layer of the Chukchi and Bering Seas. The Barents Sea turned out to be the most transparent. The differences in extinction varied more than 1.5 times. In all measurements, except for the Barents Sea, the tendency to an increase in chlorophyll fluorescence in more transparent waters was observed.

## 1 Introduction

Nowadays, global warming and the resulting changes in the environment are one of the most important problems classified by the world community as a big challenge. Global warming was predicted 40 years ago as a possible response of the natural environment to increasing anthropogenic emissions of greenhouse gases (McNutt, 2019). Recently, it has received one more very strong confirmation against the background of natural long-term climate fluctuations (George, 2019; Neukom et al., 2019). It consists in the fact that all previous climate coolings and warmings in the period of 2000 - 0 A.D. were regional, 50  while the current one covers the entire globe. At the same time, there are regions where warming occurs at a faster pace. They include the Arctic, where the air temperature increases two to three times faster than in other regions of the planet (Najafi et al. 2015; Shepherd, 2016). In this regard, natural questions arise: how does warming in the Arctic affect the air composition (Cory et al., 2014; Nomura et al., 2018; Sand et al., 2016; Travnik, 2014; Willis et al., 2018; Yasunaka et al., 2016) and how is this related to pollutants transported to the Arctic region (Arnold et.al, 2016; Evangeliou et.al, 2016; Law 55  et al., 2014; Roiger et.al, 2015)? We can answer these questions and outline a plan of measures to preserve the vulnerable nature of the Arctic only on the basis of measurements and analysis of direct relations and feedbacks between climate warming and changes in air composition (Kulmala et al., 2010). However, data in the Russian sector of the Arctic are extremely scarce. As a result, we have a situation that irreversible processes requiring an immediate response proceed in the Arctic environment, but even the estimated information about them, in particular, about the state of the atmosphere, is 60  lacking. As a result, environmental and socio-economic forecasts for the Russian sector of the Arctic are hampered because of a lack of data (Schmale et.al, 2018).

The composition of the atmosphere in the Arctic region was most often studied in the surface air layer along the coast of the Arctic Ocean (Asmi et al., 2016; Cassidy et al., 2016; Fisher et al., 2014; Giamarelou et.al, 2016; Golubyatnikov, Kazantsev, 2013; Kiselev, Reshetnikov, 2013; Langer et al., 2015; Myhre et al, 2016; Strachan et al., 2015; Willis et al., 2016). The 65  results obtained in these studies provide information on the dynamics of aerosol and gas constituents in the coastal areas and also allow estimating of the power of sources and sinks of these constituents. At the same time, the processes of gas and aerosol exchange between the water surface of the increasingly ice-free ocean and the atmosphere remain unclear. There is



no information about the vertical distribution of gas constituents and aerosol, which is very important, as was shown in the analysis of the heating of different atmospheric layers (Kylling et al., 2018; Zhuravleva et al., 2018), because air heating can

occur not only in the surface air layer, but also in the middle troposphere. This makes difficult the modeling of climatic processes and greatly complicates the forecast of environmental changes.

At the moment, there are no systematic observations of the vertical distribution of gas and aerosol components of the atmosphere. The only exception is ozone, which can be measured by serial ozone probes. The solution to the problem of monitoring the air composition in the Arctic region, as well as in other remote areas of the globe, was associated with the

development of satellite sensing systems. According to the review (Burrows, Martin, 2007), 15 types of spacecraft measuring the aerosol and gas composition of the air were already operating in 2007. However, their operation demonstrates that the satellite data do not provide the measurement accuracy necessary for practical needs yet (Tollefson, 2016), especially in the Arctic. Therefore, both the hardware and methodological components of this sensing method are to be updated. (Costantino et al., 2017; Popkin, 2017).

Flying laboratories are widely used to study the vertical distribution of gas and aerosol components outside the Russian sector of the Arctic (France et al., 2016; Leaitch et al., 2016; Quennehen et al., 2011). The airborne method for studying the vertical air composition is now recognized as a reference method (Wendisch, Brenguier, 2013), since it allows the use of high-precision in situ instruments and measurements in meteorological conditions corresponding to the objectives of the experiment with a good reference in space and time.

In the Russian sector of the Arctic, besides airborne measurements during the International Polar Year (Paris et al., 2009a) the air composition was studied only in the near-surface (near-water) air layer at the drifting stations (Nagurnyi, 2010; Nagurnyi, Makshtas, 2016) or from research vessels (Pipko et al., 2010; Semiletov et al., 2013; Yu et. al., 2015). Finally, the well-equipped Tiksi coastal observatory has appeared (Reshetnikov, Makshtas, 2012). Unfortunately, it is only one on the entire coast of the Arctic Ocean, which in Russian jurisdiction has a length of several thousand kilometers. In the last decade,

we have managed to carry out two small flight campaigns in the Russian sector of the Arctic within the framework of international projects (Antokhina et al., 2018; Antokhin et al., 2014; Petaja et al., 2021).

To fill the gap in data on the vertical distribution of gas and aerosol constituents of air over the Russian sector of the Arctic, in 2020 the experiment on sensing of the atmosphere and water surface over all seas of the Arctic Ocean (Barents, Kara, Laptev, East Siberian and Chukchi seas) was carried out from onboard the Optik Tu-134 aircraft laboratory. The Bering Sea

in the Pacific Ocean was chosen as a reference one in relation to the Arctic. It should be noted that such a large-scale experiment was conducted neither in the former USSR nor in modern Russia. This paper describes the conducted experiment, characterizes the used equipment, and provides some tentative results.





## 2 Instrumentation

The experiment was conducted onboard the Optik Tu-134 aircraft laboratory. Its basic equipment is described in (Antokhin et al., 2011). For the Arctic experiment, the basic equipment was significantly expanded. This section describes the used instruments and systems.

Figure 1 shows the arrangement of instruments and sensors both inside the aircraft and on its external elements. The measuring system consists of remote devices and devices operating on the contact principle of measurements. Their operation requires auxiliary systems, such as power supply, airlines, recording and control systems.

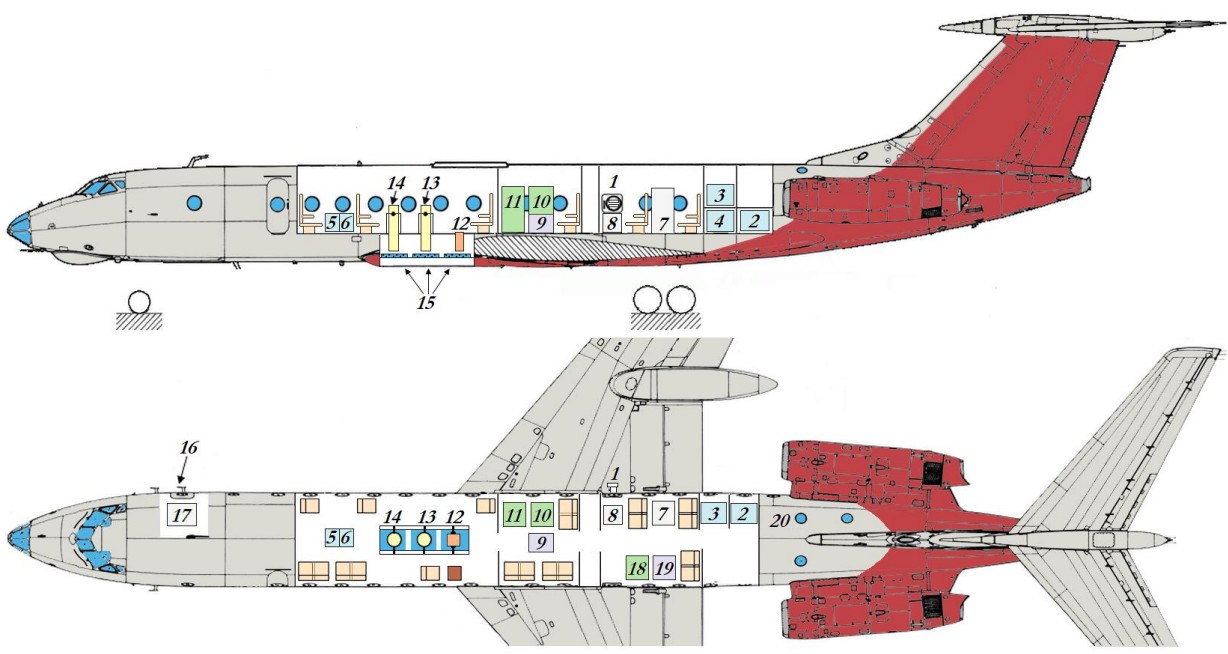

**Figure 1: Arrangement of the instrument suite on board of the OPTIK TU-134 aircraft laboratory:** *1 –* ambient air inlets and RH&T probe; *2 –* aircraft electrical power distribution unit (28 VDC); *3, 4, 5, 6 –* inverters (28 VDC/220 VAC) and UPSs (Delta RT-2K); *7 –* aerosol instrument rack: aethalometer (MDA-02) and photoelectric aerosol nephelometer (FAN-M); *8 –* aerosol instrument rack: diffusional particle sizer (DPS), optical particle counter (Grimm 1.109), filter and bioaerosol sampling suite; *9 –* navigation system (CompaNav-5.2 IAO); *10 –* gas analysis rack: $CO_2/CH_4/H_2O$ (Picarro G2301-m): *11 –* gas analysis rack: $O_3$ (TEI Model 49C), CO (TEI Model 48C); *12 –* spectroradiometer (Spectral Evolution PSR-1100F); *, 13, 14 –* aerosol lidars; *15 –* camera hatches; *16 –* ambient air inlet; *17 –* sampling unit for organic aerosol analysis; *18 –* gas analysis rack: $NO_x$ (Thermo Scientific Model 42*i*-TL); *19 –* main data acquisition system (NI PXI-1042); *20 –* GLONASS/GPS antennas.

### 2.1 Gas analysis system

To measure the concentrations of climatically significant minor gas constituents of the atmosphere, the following gas analyzers were installed onboard the Optik Tu-134 aircraft laboratory:





CO$_2$, CH$_4$, and H$_2$O - G2301-m operating based on the technology of cavity ring-down spectroscopy (CRDS, Picarro Inc., USA);

O$_3$ – Model 49C UV photometric gas analyzer (Thermo Environmental Instruments Inc., USA);

CO – Model 48C non-dispersive infrared (NDIR) correlation gas-filter analyzer (Thermo Electron Corp., USA);

NO and NO$_2$ (NO$_X$) – Model 42$i$-TL chemiluminescence gas analyzer (Thermo Fisher Scientific Inc, USA).

The integration of several of these sensors was described previously (Paris et al., 2008; Paris et al., 2010; Anthokhin et al., 2018). The gas analysis system, airlines, and electronic communications are shown schematically in Fig. 2.

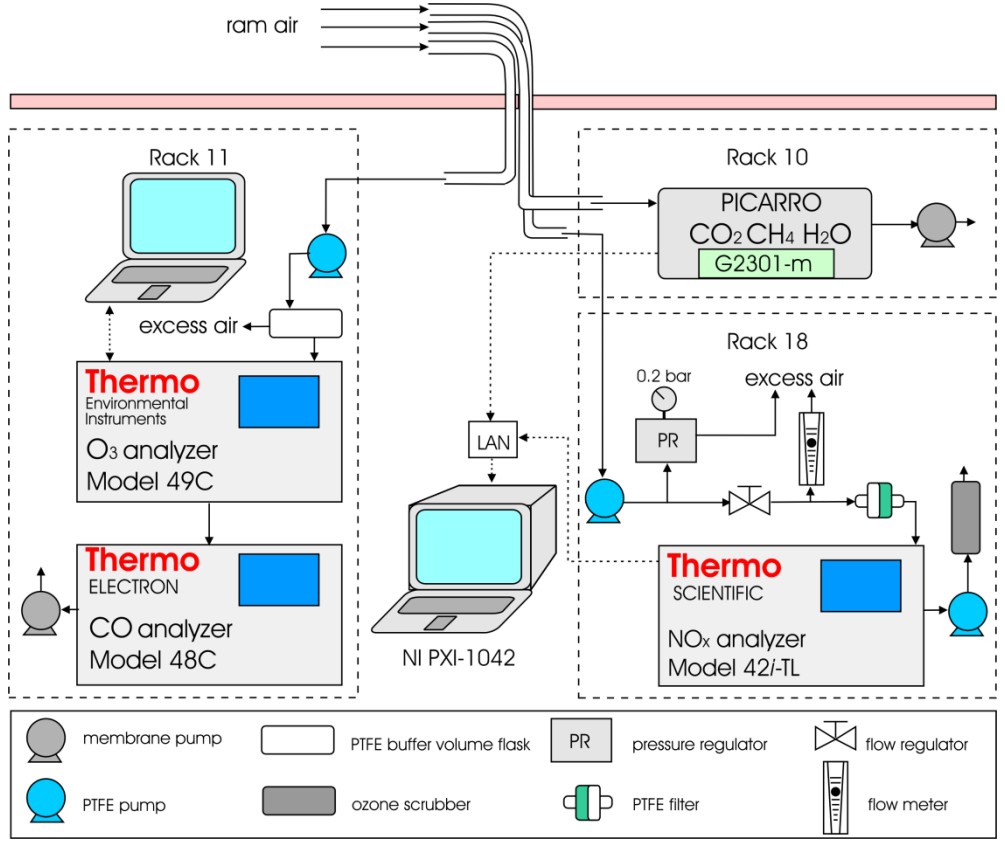


**Figure 2: Schematic diagram of the gas analysis instrument suite: solid lines – air ducts; dash lines – communication cables.**

An atmospheric air sample is supplied directly to the G2301-m gas analyzer by a Synflex 1300 1/4 combined metal-plastic tube (outer $\varnothing$ = 1/4 ″, inner $\varnothing$ = 4 mm). The N 920 APE-W diaphragm pump (KNF Neuberger GmbH) is used as a flow

stimulator, which provides the constant operating pressure at a level of 140 ± 0.1 Torr in the gas analyzer resonator up to an altitude of 10 km.



Atmospheric air samples for measurement of the concentrations of CO and chemically reactive gases $NO_x$ and $O_3$ are delivered by Masterflex™ poly-tetra-fluoroethylene (PTFE) tubes (outer $\varnothing$=6.0 mm, inner $\varnothing$=4.48 mm). In contrast to G2301-m, the Model 49C, Model 48C, and Model 42$i$-TL gas analyzers are not designed to operate at low pressures.

Therefore, air samples are supplied to them with PTFE pre-pumps: VDE 0530 (KNF Neuberger GmbH) to Model 49C and Model 48C; and LABOPORT® N820 FT.18 (KNF Neuberger GmbH) to Model 42$i$-TL.

Technical characteristics of the gas analysis system are given in Table 1.

Table 1.

Technical characteristics of the gas analyzers

| Gas analyzer | Gas | Range | Error | Time constant |
|---|---|---|---|---|
| | $CO_2$, ppm | 0...1000 | < 0.2 ppm | 1 s |
| G2301-m | $CH_4$, ppm | 0...20 | < 0.0015 ppm | 1 s |
| | $H_2O$, ppm | 0...70000 | < 150 ppm | 1 s |
| Model 49C | $O_3$, ppm | 0...200 | ± 0.001 ppm | 1 s |
| Model 48C | CO, ppm | 0...10000 | ± < 1% | 4 s |
| Model 42$i$-TL | $NO/NO_2/NO_X$, ppm | 0...0.5 | ± 0.0004 | 10 s |


## 2.2 Aerosol system

The aerosol system consists of several devices for determination of the disperse and chemical composition of particles, as well as their optical characteristics.

### 2.2.1 Devices for measurement of the particle size distribution

To study the vertical structure of the aerosol particle size distribution, two types of devices were installed onboard the Optik Tu-134 aircraft laboratory: diffusional particle sizer (DPS), which allows retrieving the nanoaerosol particle number distribution in the size range from 3 to 200 nm in 20 size intervals, and the Grimm Model 1.109 aerosol laser spectrometer (Grimm Aerosol Technik GmbH & Co., Germany) designed for measurement of the aerosol particle number density in the size range from 0.25 to 32.0 μm (Fig. 3) (Arshinov et al., 2007; Paris et al., 2009b).



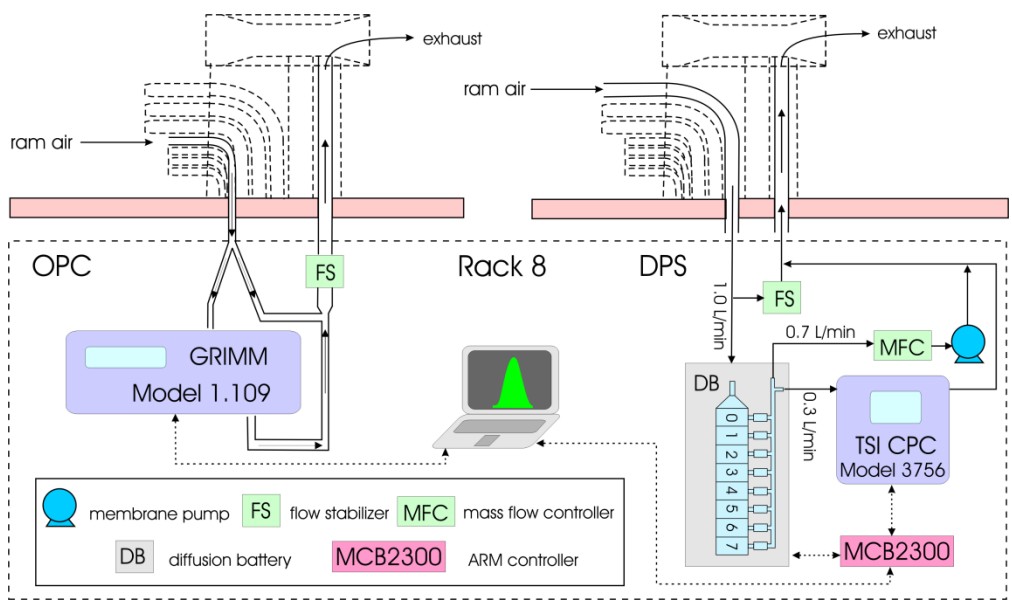


**Figure 3: Schematic diagram of the instrument suite for particle counting and sizing: solid lines – air ducts; dash lines – communication cables.**

DPS consists of an 8-section diffusion battery (DB) of mesh type (manufactured by the Institute of Chemical Kinetics and Combustion SB RAS) (Ankilov et al.,2002; Reischl et al.,1991) and CPC Model 3756 condensation particle counter (TSI

inc., USA). The particle size separation principle is based on the size dependence of the diffusion coefficient of nanoparticles. Particles of different sizes have different deposition rates when passing through porous media (smaller particles are removed from the air flow faster). Thus, the coefficient of passage of particles through such a medium bears information about the particle size. The particle number density is measured sequentially at the exit of each DB section with the CPC Model 3756 condensation particle counter. After every scan of all DB sections, the size spectrum is retrieved by the

algorithm developed by A.N. Ankilov and S.I. Eremenko (Eremenko, Ankilov, 1995), which was chosen by TSI as the basic software for DPS diffusion classifier (Knutson,1999). The use of the WCPC 3756 condensation particle counter having the response time <2 s has allowed us to obtain the entire nanoparticle size distribution for 80 s.

The Grimm Model 1.109 laser aerosol spectrometer allows the aerosol particle number density to be measured in 31 size intervals: 0.25; 0.28; 0.3; 0.35; 0.4; 0.45; 0.5; 0.58; 0.65; 0.7; 0.8; 1.0; 1.3; 1.6; 2; 2.5; 3; 3.5; 4; 5; 6.5; 7.5; 8.5; 10; 12.5; 15;

17.5; 20; 25; 30; 32 μm. Its operating principle is based on the dependence of the scattered radiation intensity on the particle size. The pulse repetition frequency at the known air flow rate makes it possible to determine the number concentration of particles in the air.

Taken together, both spectrometers form a single aerosol system capable to cover the size range from 0.003 to 32 μm with a good resolution.





The connection of the above devices to the inlet and outlet branch pipes of the air intake is shown schematically in Fig. 3. An inlet pipe ∅=7.5 mm (outlet ∅=16 mm) is used to deliver aerosol particles in the size range of 0.3-20 μm, and nanoaerosols in the size range of 3-200 nm are measured with inlet and outlet ∅=11 and ∅=28 mm, respectively. The supply lines have a bypass channel to avoid excessive pressure drop between the air inlets and outlets of the devices and to minimize the diffusion losses of nanoparticles in the supply lines. To diminish the effect of electrostatic deposition of aerosols, electrically

conductive silicone tubes were chosen as supply lines. To reduce the inertial losses of large aerosol particles, the Grimm Model 1.109 aerosol spectrometer is placed as close as possible to the intake device on aspiration rack *8* (Fig. 1). Technical characteristics of the diffusion and laser aerosol spectrometers are given in Table 2.

Table 2.

Technical characteristics of the aerosol system (disperse composition)

| Analyzer | Channels, concentration | Range | Error | Time constant |
|---|---|---|---|---|
| Diffusional particle sizer | $D_p$, nm (20 channels) | 3...200 | − | 80 s |
| | $N$, cm$^{-3}$ | 0...500000 | ± 10% | |
| GRIMM | $D_p$, μm (31 channels) | 0,25...32 | − | 6 s |
| Model 1.109 | $N$, cm$^{-3}$ | 0...2000 | ± 3% | |


### 2.2.2 Equipment for measurement of Black Carbon and aerosol scattering

To measure the content of absorbing substance (black carbon), two devices were used onboard the aircraft laboratory, namely, the AE33 serial aethalometer and MDA-2 developed in IAO SB RAS. In its operating principle, MDA-2 is analogous to the device of the aethalometer type developed by Hansen et al. (Hansen et al., 1984). Its operation is based on

continuous measurements of the diffuse attenuation of radiation by a layer of aerosol particles directly in the process of their deposition on the filter from the pumped air. In this case, the value of the recorded diffuse attenuation of light by the layer of particles is directly proportional to the Black Carbon (BC) surface concentration on the filter and, consequently, to its mass concentration in the air. In the MDA-2 aethalometer, the ambient air flow enters the optical cell of the device through a hose with an inner diameter of 8 mm and a length of about 2 m, the average intake capacity is 4 liter/min. With these parameters

of the airline, coarse aerosol particles larger than 1 μm are mainly deposited in the inlet hose. Thus, the aethalometer mostly records the content of the BC component only in the submicron aerosol fraction. The block diagram of the optical BC meter (aethalometer) is shown in Fig. 4a.



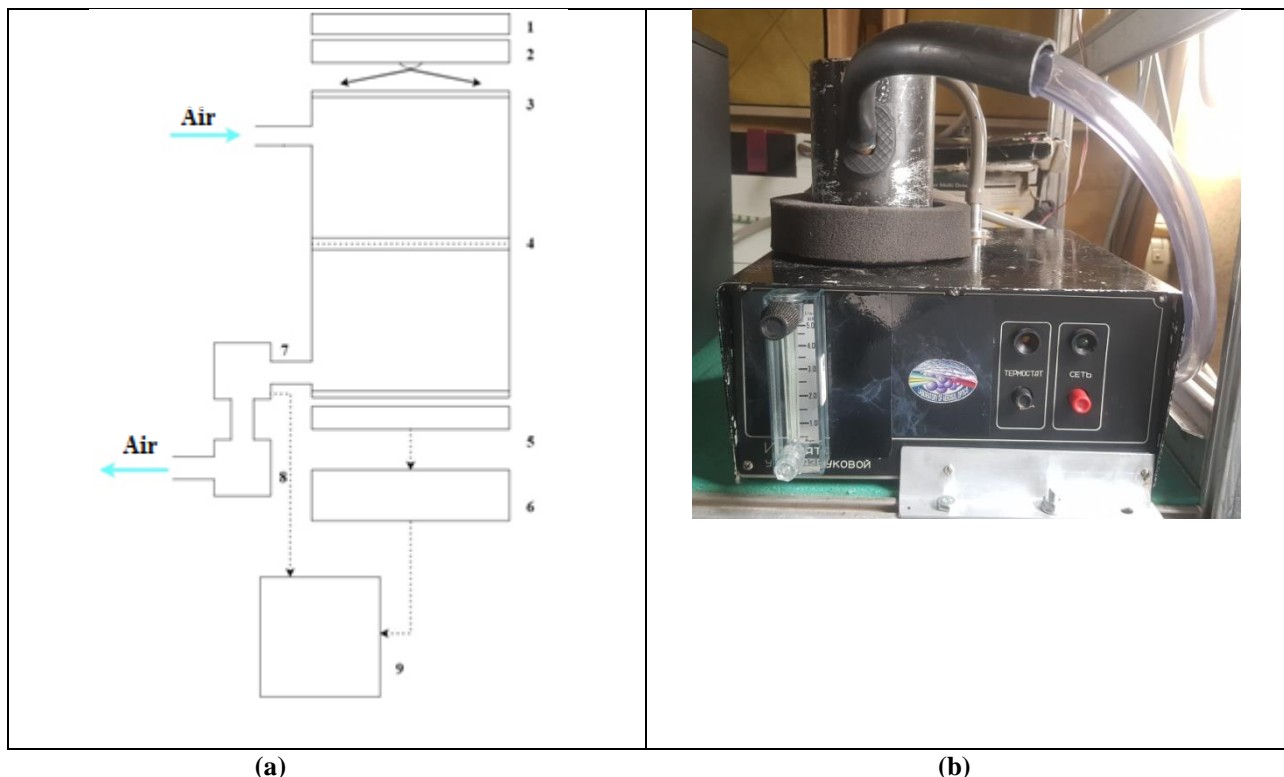

| (a) | (b) |

**Figure 4: Block diagram (a) and photo (b) of MDA-2: (1) thermostat, (2) radiation source, (3) optical cell, (4) AFA-HP-20 aerosol filter, (5) photodetector, (6) controller, (7) flow meter, (8) air intake pump, (9) computer.**

The main structural elements of the optoelectronic unit are a radiation source, an optical cell, and a unit for amplification and registration of signals. Two thermostated RYGB LZ4-20MA10 LEDs are used as a radiation source. Two TSL 237 light-frequency converters served as photodetectors. The LEDs are connected in series so that the current through them is the same. The heating is controlled by two additional temperature sensors: one at the source and another at the receiver. Control and information reading are carried out with the ATMega-8 controller. The RS-232 serial port serves for communication with the computer. A rotameter of 5 liter/min and an AWM43600 sensor operating in the range of 0-6 liter/min are installed to control the flow. In total, the optical range of the light sources is 460, 530, 590, 630 nm. The test air is pumped through the filter section in the measuring channel, and a layer of BC particles accumulates on it, causing the filter to blacken. The filter section in the reference channel is not "exposed" during measurements and remains clean. The signal converter operates by the differential scheme and continuously measures the signal difference between the measuring and reference channels during the filter blackening with the deposition of particles. The number concentration of BC particles in the air is calculated by the software, which also controls the device operation in the automatic mode.

The absolute calibration of the aethalometer was based on the comparison of the data of synchronous optical and gravimetric measurements of BC aerosol (Baklanov, et al.,1998). During the calibration, BC particles with a size of 50–200 nm were



used, which were formed in the pyrolysis of butanol vapors in a nitrogen atmosphere at a temperature of 1150°C.

Before the start of the flights, the AE33 and MDA-2 were intercalibrated. The intercalibration showed that the concentrations measured by these two devices differ on average by no more than 5-7% at different generation modes used to construct the calibration characteristics of the device.

*FAN-M nephelometer*

The scattering properties of the atmosphere were studied with a FAN-M automated nephelometer designed to measure the coefficient of directional scattering of radiation at the scattering angle of 45 $\mu_d$ ($\varphi = 45°$) (Mm$^{-1}$sr$^{-1}$) at a wavelength of 0.53 μm. The volume scattering coefficient was estimated by the empirical relationship $\sigma_d = 7.3 \cdot \mu_d(45°)$ (Mm$^{-1}$) (Panchenko et al., 2019). The nephelometer is calibrated in every flight against molecular (clean) air at different altitudes. For this purpose, outboard air is passed through a set of three to four AFA filters before coming to the device. This set of filters completely

deposits the aerosol component of atmospheric air on the filtering material.

Technical characteristics of the aethalometer and nephelometer are given in Table 3.

Table 3.

Technical characteristics of the aethalometer and nephelometer

| Device/sensor | Parameter | Range | Error | Time constant |
|---|---|---|---|---|
| AE -33 aethalometer | BC, ng/m$^3$ | 10 ng/m$^3$…100 μg/m$^3$ | 100 ng/ m$^3$ | 60 s/ 1 s |
| MDA-2 aethalometer | BC, μg/ m$^3$ | 0,01…100 | 0,01 μg/ m$^3$ | 20 s |
| FAN-M nephelometer | σ, км-1 | 0.001 …2 | 5%-1 | 1 s |

### 225  2.2.3 Sampling devices

Three kinds of sampling were used in the experiment. Correspondingly, three sampling systems were made. Two of them are designed to determine the organic and inorganic components of aerosol particles. They are based on particle sampling onto filters. This method is traditional and has been described many times in the literature. These two systems differ in the number of filters and the air flow rate through the filter.

The situation with sampling of bioaerosols is more complicated. For this purpose, a separate unit was designed and manufactured. Sampling was carried out in MTS-50 impingers manufactured by Experimental Design Bureau of Fine Biological Engineering, Kirishi, Russia. The diagram of the setup for sampling onto impingers and its appearance are shown in Fig. 5.

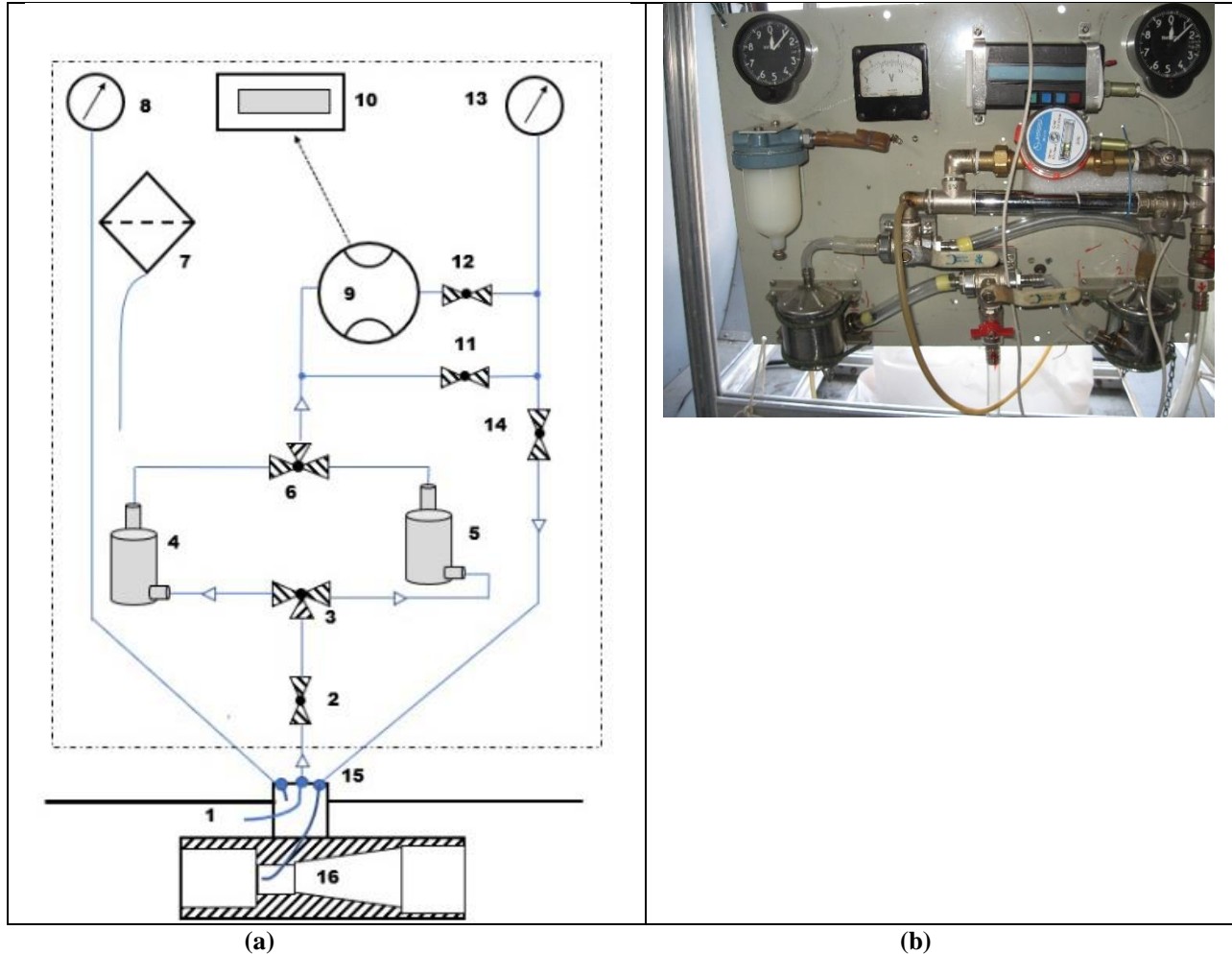

**(a)**            **(b)**

**Figure 5: Block diagram and appearance of the onboard rack for bioaerosol sampling: 1 – air intake (isokinetic sampler); 2,11,12,14 – ball valves; 3,6 – three-way ball valves; 4,5 – impingers; 7 – FV-1.6 filters; 8,13 – VD-10 aviation altimeters; 9 – non-contact flow meter sensor; 10 – flow meter digital unit; 15 – quick-disconnect connections to air intake and sensor units ; 16 – air outlet (Venturi nozzle).**

The device is made of stainless steel with a critical nozzle that ensures a constant air flow through the device at a pressure
drop of more than $4\times10^4$ Pa. The required pressure drop was provided by the pressure difference of the flow on the air intake during the aircraft movement and the pressure overboard. Particles are deposited into the liquid swirling along the device walls by the incoming flow (with a flow rate of 50 ± 5 liter/min). 50 ml of colorless Hanks solution (ICN Biomedicals) is used as the sorbing liquid. The deposition efficiency of this device for aerosols larger than 0.3 μm (the minimum size of most known bacteria) exceeded 80%, while that for particles more than 2 μm in diameter was equal to an almost constant



value of 90 ± 15%. To eliminate the loss of aerosol particles in the connecting tubes, the impingers are located near the isokinetic sample intake.

In addition, the setup for sampling the inorganic component served also for bioaerosol sampling for the following analysis. Bioaerosol was sampled onto AFA-HA-20 filters for subsequent analysis of the total protein mass on the filter and/or analysis of genetic material of various origins in a sample.

## 2.3 Lidars

During the flight campaign, the aircraft  laboratory operated two onboard lidars: LOZA–A2 and CE-372NP. The lidar data were used to retrieve optical characteristics of the atmospheric environment (aerosol) and sea water. For the atmosphere, the following profiles were measured: profile of the aerosol backscattering coefficient for wavelengths of 532 and 1064 nm, profile of the index of radiation depolarization at aerosol particles, and profile of the accumulated aerosol optical depth

(possible only when measuring in the dark). For the sea water, the measured characteristics are the extinction coefficient in water in the range 0.07-0.6 m⁻¹ at a wavelength of 532 nm and the relative concentration of phytoplankton by laser-induced fluorescence (LIF) of chlorophyll-a at a wavelength of 680 nm (the concentration can be determined at the calibration against *in-situ* measurements).

The technical characteristics of the both lidars are described in sufficient detail in the scientific literature (Ancellet

et al.,2019; Dieudonné et al.,2015; Mariage et al., 2017; Nasonov et al.,2020). Therefore, here we dwell only on the modernization of the LOZA-A2 lidar to fulfill the tasks of this large-scale experiment on sensing the sea surface. This modernization has allowed us to measure the turbidity of the upper water layer and to estimate the content of organic matter in it. The external view of the LOZA-A2 lidar transceiver onboard the aircraft is shown in Fig. 6a.

The modernization included modification of the optical scheme of the transceiver, without changing the lidar bearing

structure, and the addition of special photodetector units that allow recording LIF both in a narrow spectrum of selected wavelengths (with a central line at 685 and 740 nm) of received radiation and in a wide spectrum from visible ($\geq$ 550 nm) to near infrared ($\leq$ 1000 nm).

For this purpose, the receiving objective (AL1 in Fig. 6b) was replaced in the channel for recording of lidar signals at a wavelength of 1064 nm. The newly developed achromatic double-lens objective with a light aperture A = 110 mm and a

focal length f = 550 mm allows eliminating chromatic aberrations and correctly recording signals in a wide spectrum of wavelengths from the visible to the near-IR region. Optical units, which separate the received radiation at the LIF wavelengths from the axial channel recording the elastic scattering signal at a wavelength of 1064 nm, are added to the main channel. A BS dichroic mirror (DMLP900R, Thorlabs GmbH, Germany) in Fig. 6 set at an angle of 45° to the main optical axis splits the received spectrum, reflecting 92% in the range 400-900 nm and transmitting 90% in the range 930-1300 nm.

Metal mirror M with silver coating (PFR10-P01, Thorlabs GmbH, Germany) diverts the reflected light flux to the FF fluorescent filter (86-988 - OD 6, central wavelength of 680 nm, bandwidth of 20 nm, Edmund Optics, USA). Then the cut-out band at these wavelengths is focused by lens L onto the photocathode of the recording PM 680 photodetector (H11526-


20-NF, Hamamatsu Photonics K.K., Japan). The fluorescent filter can be quickly replaced with a wider filter (87-757 - OD 6, central wavelength of 700 nm, bandwidth of 75 nm, Edmund Optics, USA). This allows us to record LIF signals from
different photosynthetic pigments of the phytoplankton community at fixed wavelengths by the high-sensitivity gated photodetector in the analog mode.

The 680-nm fluorescent channel is made as a separate unit and can be replaced without disturbing the optical alignment of the entire lidar. As a result, the optical unit for simultaneous recording of a wide range of LIF wavelengths in the received radiation can be quickly installed in the lidar. The optical scheme of this unit is shown in Fig. 6 (top panel). In this unit, an
FA optical matching adapter is set instead of the FF fluorescent filter and the PM 680 photodetector. The double chromatic lens focuses the received radiation at the end face of fiber F (optical fiber diameter ≤ 1 mm). The optical fiber from the lidar is connected to the PMA multichannel spectrum analyzer (PMA-12, C10027-01, Hamamatsu Photonics K.K., Japan). An IF cutoff filter (FELH0550 long-wavelength filter, cutoff wavelength of 550 nm, Thorlabs, Germany) is additionally installed in this unit. The IF filter blocks the second-harmonic laser wavelength at 532 nm. The use of the PMA-12 spectrometer for LIF
analysis allows us to record the spectrum of decaying fluorescence of *chlorophyll-a* contained in all plants in the dark, as well as to analyze the spectra of radiation reflected from the underlying surface.

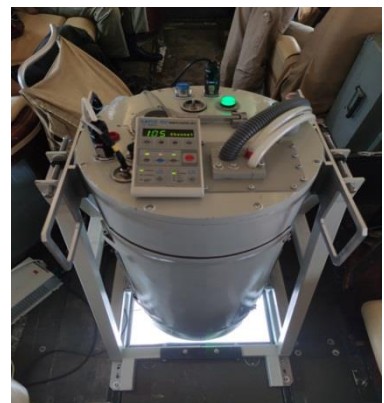
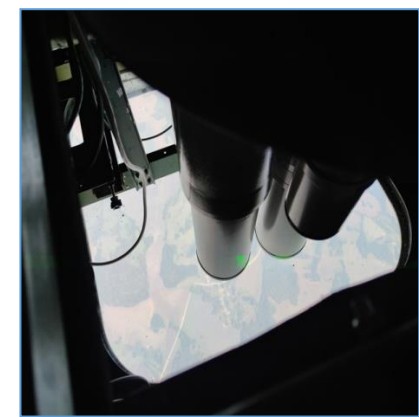

**(a)**



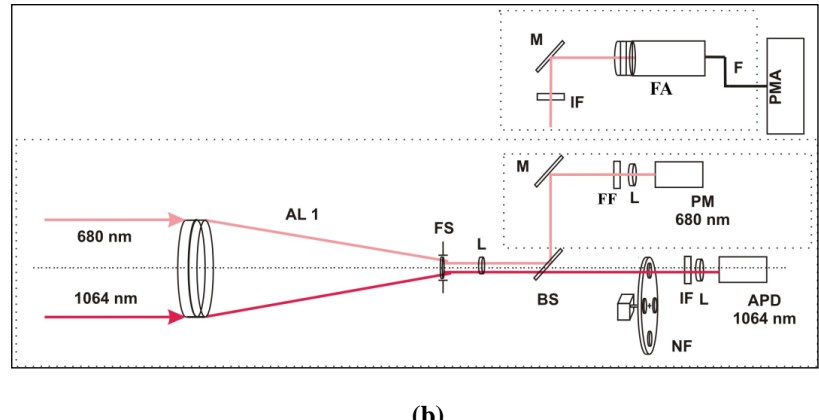


**(b)**

**Figure 6: Photo (right) of the 680 fluorescent channel unit built in the lidar (a) and the optical arrangement of the lidar with fluorescent channels (b).**

### 2.4 PSR-1100f spectroradiometer

The PSR-1100f spectroradiometer (hereinafter referred to as PSR-1100f) allows measurements of incoming/absorbed
radiation in the visible and near-IR spectral regions. The receiving unit of the spectroradiometer is a cross-correlation cell of the Čzerny-Turner configuration, in which a rifled diffraction grating is used as a dispersing element. The light flux enters the spectrometer and is collimated before being reflected from the grating and refocused onto the PDA detector. The detector is an array of 512 elements covering the spectral range 320-1100 nm. It has a built-in microprocessor that controls the data acquisition by the detector array and the interface with the host computer, as well as provides data storage. The stored data
consists of the calibration data (in flash memory), current dark and reference scan data (in RAM memory), and current spectrum scan data (also in RAM memory). The microprocessor also performs mathematical operations on the scanned data, such as dark signal subtraction and auto integration adjustment. The PSR-1100f has a built-in shutter for dark measurements.

Technical characteristics of PSR-1100f:

Spectral range – 320–1100 nm;

Nominal spectral resolution – <=3.0 nm;

Spectral bandwidth – 1.5 nm;

Accuracy of wavelength determination – 0.5 nm;

Wavelength reproducibility – 0.1 nm;

Integration time – 8-2000 ms;

Built-in memory – up to 2500 spectra.

The measurements are carried out in the flying aircraft upon the protective shutters open. These shutters protect the hatches from adverse factors during the aircraft parking, takeoff and landing. In flight, the PSR-1100f is controlled by the software installed on the host computer via Bluetooth interface.



## 2.5 CompaNav-5.2 IAO integrated inertial system

The CompaNav-5.2 IAO navigation system was developed and created specially for flights in the Arctic in addition to the existing navigation system of the Tu-134 aircraft. The CompaNav-5.2 IAO strapdown inertial navigation system (INS) is designed to determine the coordinates of location, motion parameters, and orientation angles of an aerial or ground vehicle. INS is based on Russian-made fiber-optic gyroscopes and MEMS accelerometers. CompaNav-5.2 IAO includes a GPS/GLONASS Ublox NEO-8M receiver as an additional source of navigation information. The INS block diagram is

shown in Fig. 7, and technical characteristics are given in Table 4.

Table 4.

Accuracy characteristics

| Parameter | Integrated mode INS/SNS/SVS |
|---|---|
| Horizontal coordinates | 6 m |
| Ground speed | 0,1 m/s |
| Vertical speed | 0.15 m/s |
| Orientation angles (roll, pitch) | 0.07° |
| Course (after 10-min initialization from SNS) | 0.2° |
| Height | 4 m |
| Air speed | 1.5 m/s |

CompaNav-5.2 IAO on aerial vehicles operates in two modes: integrated inertial-satellite mode (in the presence of a high-quality signal) or autonomous mode with correction from the air signal system (in the absence or uncertain reception of a satellite signal). The built-in GLONASS/GPS receiver is used as a source of satellite signal. CompaNav-5.2 IAO INS provides the continuous navigation support in case of prolonged (1 hour or more) loss of satellite signal. The advantages of the CompaNav-5.2 IAO INS over the previously used navigation system are compactness (CompaNav-5.2 IAO weight of

6 kg), positioning accuracy (3 decimal places), low power consumption, and the number of motion and positioning parameters determined (35 versus 5).







Figure 7: Block diagram of the CompaNav-5.2 IOA navigation system.



### 2.6 Meteorological system

The meteorological system of the aircraft laboratory allows measuring the following parameters: air temperature (°C), temperature of complete deceleration (°C), relative air humidity (%), and atmospheric pressure (mm Hg) (Anokhin et al.,2011 Antokhin et al., 2012).

Upon CompaNav-5.2 IAO INS was installed at the Optik aircraft laboratory, it became possible to determine the wind speed (m/s) through calculation and wind direction (°). The wind speed and direction are determined with the use of navigation
triangle.

## 3 Flight experiment over the Russian sector of the Arctic

### 3.1 General characteristic of the experiment

Flights in the Russian sector of the Arctic were carried out from September 4 to 17 of 2020. In this period, Russia slightly relaxed restrictions related to the coronavirus pandemic, although the pandemic certainly affected the preparation of the
experiment. In particular, there were difficulties with flight planning. Unfortunately, the network of large airfields on the Arctic coast is rare and far between. Some airports had 14-day observation requirements. In this situation, most flights took place without alternate aerodromes with the so-called "decision line," when the crew in the middle of the route had to decide to continue the flight or to return to the departure point.

### 3.2 Flight routes

The route map of the experiment is shown in Fig. 8. Vertical profiles were collected from the minimum permissible heights (typmically 500 m above ground level and 200 m above seas) to the upper troposphere. The flights were carried out over all seas of the Russian sector of the Arctic and coastal territories. The minimum height was 200 m above the sea and 500 m above land. Horizontal sections of the flight were performed at three altitudes: 200, 5000, and 9000 m. In these sections, aerosol samples were taken and the spatial homogeneity in the distribution of gas components and meteorological
parameters was investigated. Four vertical profiles were measured over each of the seas.





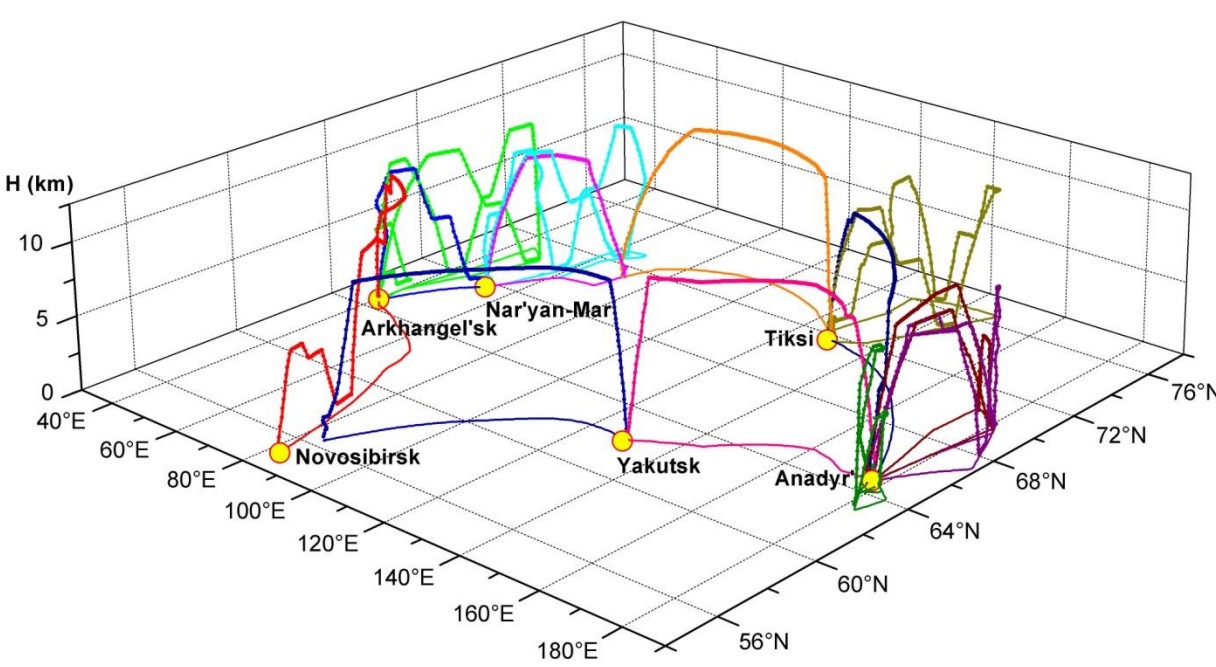

**Figure 8: Schematic map of the routes of the Optik Tu-134 aircraft laboratory in the regions of the Russian Arctic.**

The experiment started with the Novosibirsk - Arkhangelsk flight on September 4, 2020. The sensing over the Barents Sea was carried out on the same day. The Arkhangelsk - Naryan-Mar flight took place on September 6, 2020. On the same day, a

flight to the water area of the Kara Sea was made. On September 7, a flight was carried out on the Naryan-Mar - Sabetta - Tiksi route. The flight over the waters of the Laptev Sea was performed on September 9. The Tiksi - Anadyr flight took place on September 11, 2020. Initially, we planned to carry out flights over the East Siberian and Chukchi Seas from the Pevek airport. However, this airport was closed for repairs during the experiment. Therefore, the flights started from Anadyr, which reduced the possibility of covering a significant part of the sea area due to the large flight distance. The first flight was

made to the Chukchi Sea on September 15, 2020, as the clouds over the East Siberian Sea were below 150 m. Flights over the East Siberian and Bering Seas were carried out on September 16. The experiment ended with the Anadyr - Yakutsk - Tomsk flight on September 17, 2020.

### 3.3 Synoptic situation in the flight area

The flight from Novosibirsk to Arkhangelsk on September 4 took place against the background of an anticyclone with the

center located northeast of Kazan. In the take-off and landing areas, the weather was determined by the eastern and northwestern peripheral parts of this anticyclone, respectively. During the flight over the Barents Sea, the weather conditions





were determined by the transition zone between the cyclone centered over the Norwegian Sea and the anticyclone centered northeast of Kazan (Fig. 9a). The temperature and humidity conditions of the studied region were determined by a polar air mass. A warm Arctic front passed over the region at about 15:00 LT.

During the flight from Arkhangelsk to Naryan-Mar on September 6, the weather conditions were caused by a multicenter cyclone located over the North Atlantic Ocean and the seas of the Arctic Ocean and an anticyclone with centers north of Orenburg and east of Khatanga. The temperature and humidity conditions on this day were determined by a moderate air mass. During the flight from Naryan-Mar to the Kara Sea, the synoptic conditions remained similar to those observed during the previous flight (Fig. 9b).

On September 7, the Naryan-Mar - Sabetta - Tiksi flight took place. Near the ground, the weather was caused by the southeastern part of the cyclone with the main center located near Iceland. The humidity and temperature conditions were determined by a moderate air mass, and only at the end of the flight route a warm Arctic front was observed. The weather from Sabetta to Taimyr was determined by the same cyclone and then by a low-gradient high-pressure baric field. Arctic air mass was observed along the entire route.

On September 09, measurements were conducted over the Laptev Sea. The weather conditions in the area of take-off and landing were determined by the northwestern periphery of the anticyclone and further north by the powerful cyclone occupying most of the Arctic (Fig. 9c).

During the flight from Tiksi to Anadyr on September 11, a low-gradient high-pressure field was observed along the entire route. The temperature and humidity conditions were determined by the Arctic air mass.

On September 15, during sensing over the Chukchi Sea, the weather was determined by the northern part of the cyclone centered over the Bering Sea (Fig. 9d).

On September 16, measurements were taken over the East Siberian Sea. At this day, the weather near the surface was determined by the axis of elongation of the saddle formed by the cyclones centered over the Primorsky Territory and the Bering Sea and the anticyclones over Yakutia and the Beaufort Sea (Fig. 9e).

On the same day, during sensing over the Bering Sea, the weather conditions were determined by the cyclone centered over Cape Dezhnev (Fig. 9f).

The return Anadyr - Yakutsk - Tomsk flight took place on September 17. The weather at the section from Anadyr to Markovo was determined by the northeastern part of the cyclone centered over the Chukotka Peninsula. Further, up to Podkamennaya Tunguska, an anticyclone was observed with the center in the Vilyuisk region (Fig. 9g). Near Bratsk, a zone

of the warm Arctic front associated with the cyclone centered west of Omsk was observed (Fig. 9h). This cyclone with two frontal systems determined the weather conditions till the end of the flight.





(a) - 04.09.2020 – 12 UTC

(b) - 06.09.2020 – 12 UTC

(c) - 09.09.2020 – 06 UTC

(d) - 15.09.2020 – 00 UTC





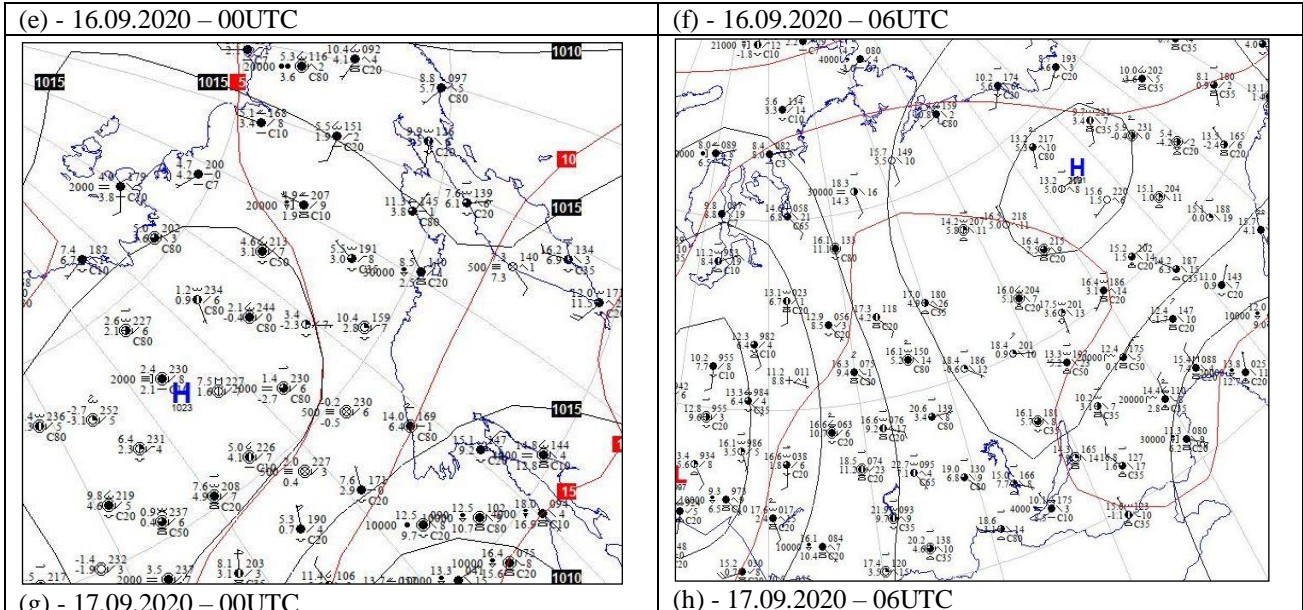

**Figure 9: Surface weather maps for September 04-17, 2020. UTC time and date are given in the appropriate box.**

## 4 Results and discussion

In the experiment, we have obtained a huge amount of data. This section presents tentative results of processing of the

measured data.

### 4.1 Gas composition

Since global warming is mainly associated with a change in the gas composition of the atmosphere, we start our consideration from the analysis of the vertical distribution of greenhouse gases.

It is known (Watson et al., 2020) that the ocean absorbs up to 25% of carbon dioxide additionally emitted by anthropogenic

activities. The high absorption capacity of the ocean may be reflected in our experiment. It can be seen from Fig. 10a that the carbon dioxide concentration decreases noticeably in the lower part of the profiles related to the atmospheric boundary layer (ABL). The $CO_2$ content in ABL over the sea areas in the measurement period was lower than that over coastal areas, as can be seen from the comparison of parts *a* and *b* in Fig.10.

The difference in $CO_2$ sinks over different seas is clearly visible. If we estimate it as the difference between concentrations at

the ABL top and at a level of 200 meters from the water surface, then it is the largest over the Barents and Kara Seas. Here the vertical gradient reaches 14 ppm. For the Laptev and Chukchi seas, it is much smaller and equal to 4 ppm. Over the East Siberian Sea, the $CO_2$ content even grows in the boundary layer. However, this is due to the transfer from the continent. To check this, initially unplanned sensing was carried out over the Bering Sea. It confirmed this conclusion.





The experimentally obtained values of the $CO_2$ concentration over the Arctic seas are significantly higher than those
published in (Cassidy et al., 2016; O'Shea et al., 2014; Pipko et al., 2010; Semiletov et al., 2013; Strachan et al., 2015). This
is understandable, since the concentration of carbon dioxide in the atmosphere is increasing all over the world.

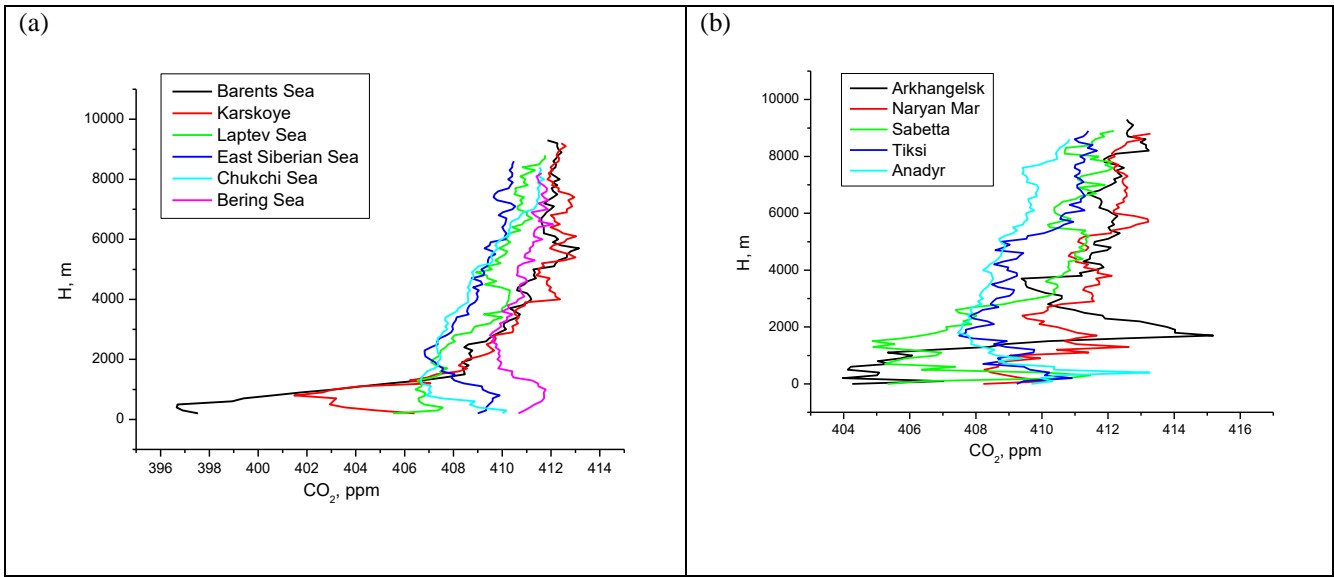

**Figure 10: Vertical distribution of carbon dioxide over sea (a) and coastal areas (b).**

The next contributor to radiation forcing is methane. As a long lived greenhouse gas, its effect do not depend on location.
However several, poorly known sources are located in the Arctic. The importance of studying the dynamics of these Arctic
sources to the atmosphere has increased sharply in the recent years in connection with the degradation of permafrost. The
fact of the presence of high $CH_4$ concentrations over this region was established both in near-surface measurements (Ivakhov
et al., 2020; Poddubny, et al., 2020) and in the entire tropospheric thickness according to satellite data (Bogoyavlensky et al.,
2020; Sitnov, Mokhov, 2018). However, opinions regarding the mechanisms of formation of this phenomenon differ.

The methane inflow directly from the ocean surface into the atmosphere was analyzed by numerical simulation on the basis
of direct measurements. The analysis showed that this amount of $CH_4$ is insufficient for the formation of the observed
concentrations (Berchet et al., 2016; Berchet et al., 2020; Li et al., 2020).

Extra methane may come from the decomposition of gas hydrates on the ocean floor. This phenomenon was observed, in
particular, during measurements from sea vessels (Sapart et al., 2017; Shakhova et al.,2010; Shakhova et al., 2015; Steinbach
et al., 2021). The scale of the threat from the decomposition of gas hydrates has been largely debated (Thornton et al., 2021;
Nisbet et al., 2020; You et al., 2019, Weber et al., 2019; Etiope et al., 2020). The contribution from this source in the
synthesis of (Saunois et al., 2016) is thought to represent less than 1% of annual emissions.

Methane can be transported to the Arctic Ocean from the surrounding land (Bozem et al.,2019; Makosko, Matesheva, 2020)
or produced in situ in the water column. The Arctic receives carbon-rich riverine influx as well as water masses from the
more southern oceans, which carry additional microbes and bacteria (Alekseev et al., 2019). They, in turn, can produce some



additional methane, which is then released into the atmosphere (Babin, 2020; Lewis et al., 2020). The process of methane (and other gases) transport to the Arctic turned out to be so significant and poorly understood that the German Aerospace Agency has organized a special HALO-AC program, in which three flying laboratories will simultaneously study this process at different altitudes (Wendisch et al.,2021).

The other possible way is the emission from aquatic ecosystems including coastal ecosystems, lakes and wetlands, where the
warming climate leads to intense decomposition of permafrost being a huge reserve of organic matter, which is processed by anaerobic microorganisms into methane or carbon dioxide (Anisimov et al., 2020; Elder et al., 2020; Marushchak et al., 2016). The reserves of organic matter are so large that in (Brouillette, 2021) they are called a "buried carbon bomb," which can explode with further warming in the region. In addition to the methane release from the soil as a result of permafrost thawing, thermokarst lakes are formed in such areas. The methane release from such lakes is an order of magnitude more
intense than that from other sources and can reach hundreds of grams per square meter a year (Bogoyavlensky et al., 2019; Jammet et al., 2015; Tan et al., 2016). According to (Walter et al., 2014), these lakes in flat regions can occupy from 10 to 30% of the territory. Therefore, this source is quite comparable to others in power.

Our measurements show that, as expected, elevated methane concentrations are observed over the Arctic seas (Fig. 11).

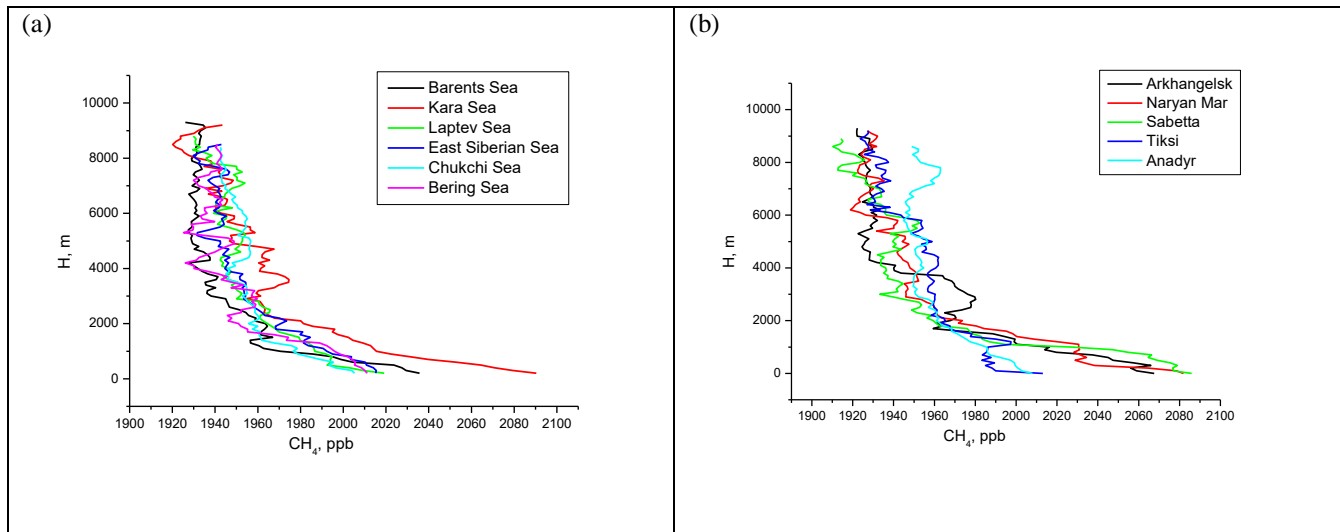

**Figure 11: The same as in Fig. 10 but for methane.**

The recently published review on the methane budget in the atmosphere (Saunois et al., 2020) although clarifies the orders of magnitude of individual sources for the planet as a whole, but does not answer the question on the reasons for the accelerated increase in the $CH_4$ concentration over the Arctic compared to other latitudes.

In contrast to $CO_2$, mixing ratios of $CH_4$ over the Arctic seas decreased with height. Thus, the lowest carbon dioxide concentrations were observed in the near-water layer over the Kara (406 ppm) and Barents Seas (399 ppm). The methane
concentrations there, to the contrary, were the highest and equal to 2092 and 2071 ppb, respectively (Fig. 11a). Over the other studied seas, the methane content in near-water layer was nearly identical and equal to 2018-2022 ppb. In a free





atmosphere, the methane concentrations over all the seas differ slightly and fall in the range of 1920-1960 ppb. To be noted is the fact that the $CH_4$ concentration of air in the coastal areas is comparable (Fig 11b) with the adjacent water areas (Fig.11a). In our opinion, this indicates the methane transport from land to sea. This conclusion is, in principle, clear from

the above synoptic maps (Fig. 9) and follows from the constructed back trajectories (Fig. 12).

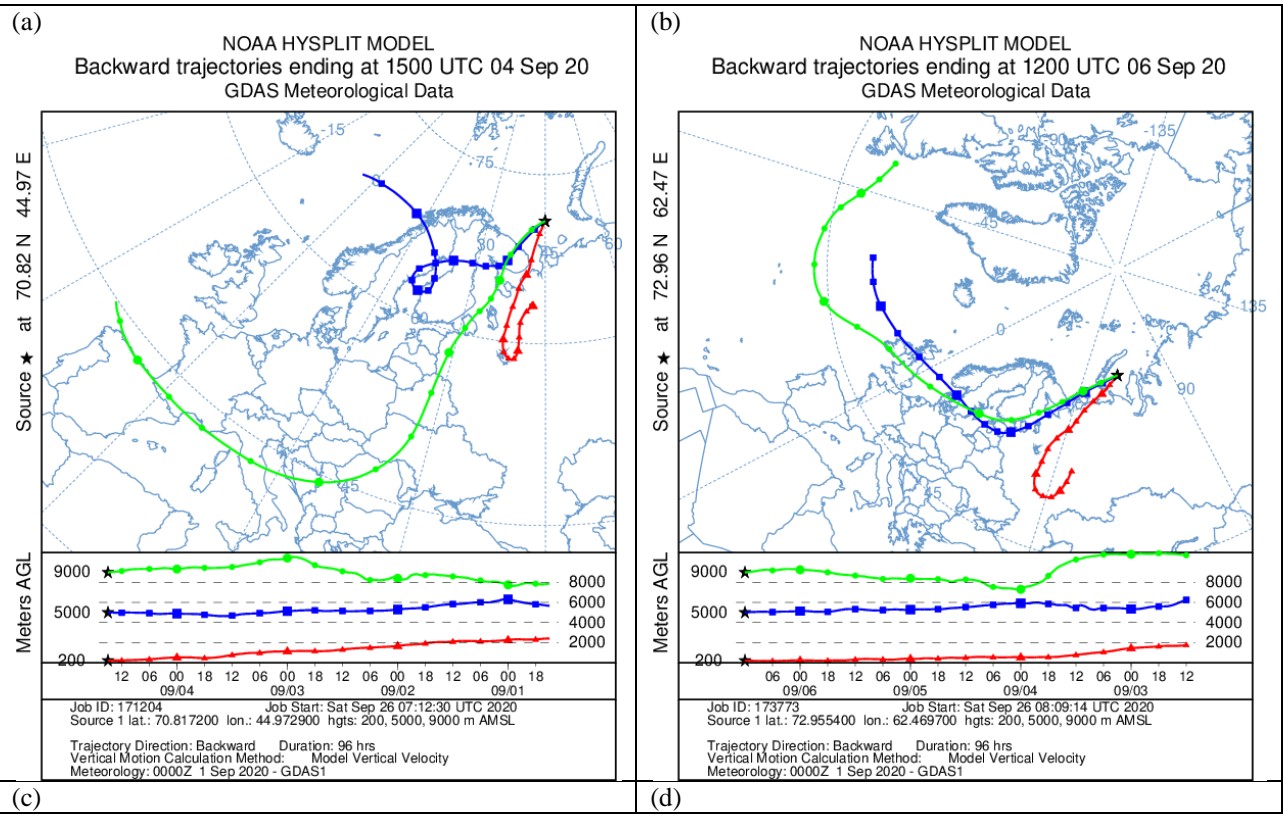





**Figure 12: Back trajectories for the period of the experiment.**


It can be seen that for the Barents and Kara Seas, where the highest $CH_4$ concentrations are observed in the near-water layer, the transport trajectories starts from the continent. Methane is most probably released by thermokarst lakes, which are

abundant in the coastal tundra as can be judged from the photograph showing the underlying surface between Arkhangelsk and Naryan-Mar (Fig. 13).

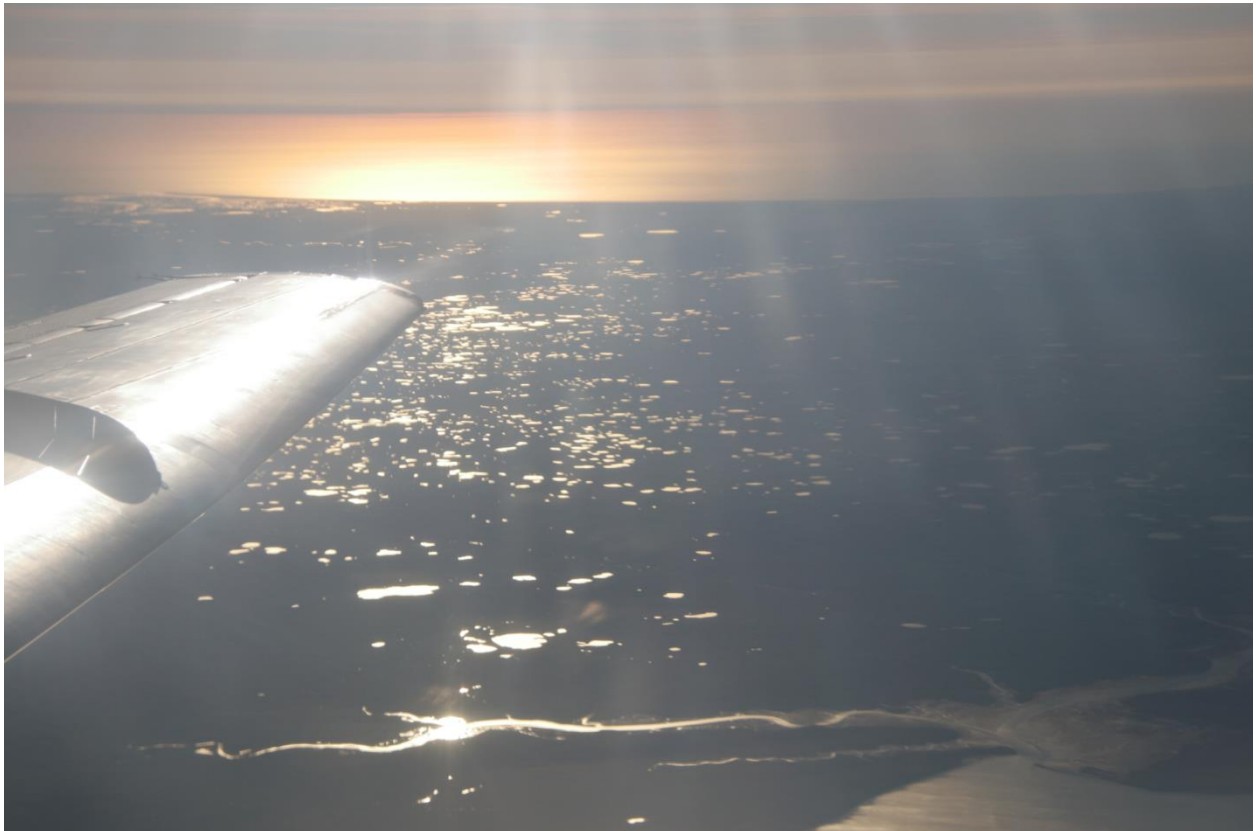

**Figure 13: A photo of the Arctic tundra with formed lakes.**

There is one more point worth dwelling on. The presence of a huge methane reservoir at the bottom of the Laptev Sea, which

could suddenly release a large amount of this gas into the atmosphere, was recently announced (Szakal, 2021). Our route over the Laptev Sea, in particular, at an altitude of 200 m almost completely coincided with the areas identified in (Szakal, 2021) (Fig. 8). However, we did not detect elevated concentrations of $CH_4$ here.

Tropospheric ozone is the fourth largest contributor to total radiative forcing (WMO, 2019). Our previous studies have shown that its formation in the lower atmosphere of the Arctic regions is insignificant (Antokhina et al., 2018; Antokhin et

al., 2014). This experiment confirms this conclusion (Fig. 14). Data for coastal areas are not shown here, as no differences with water areas were found.





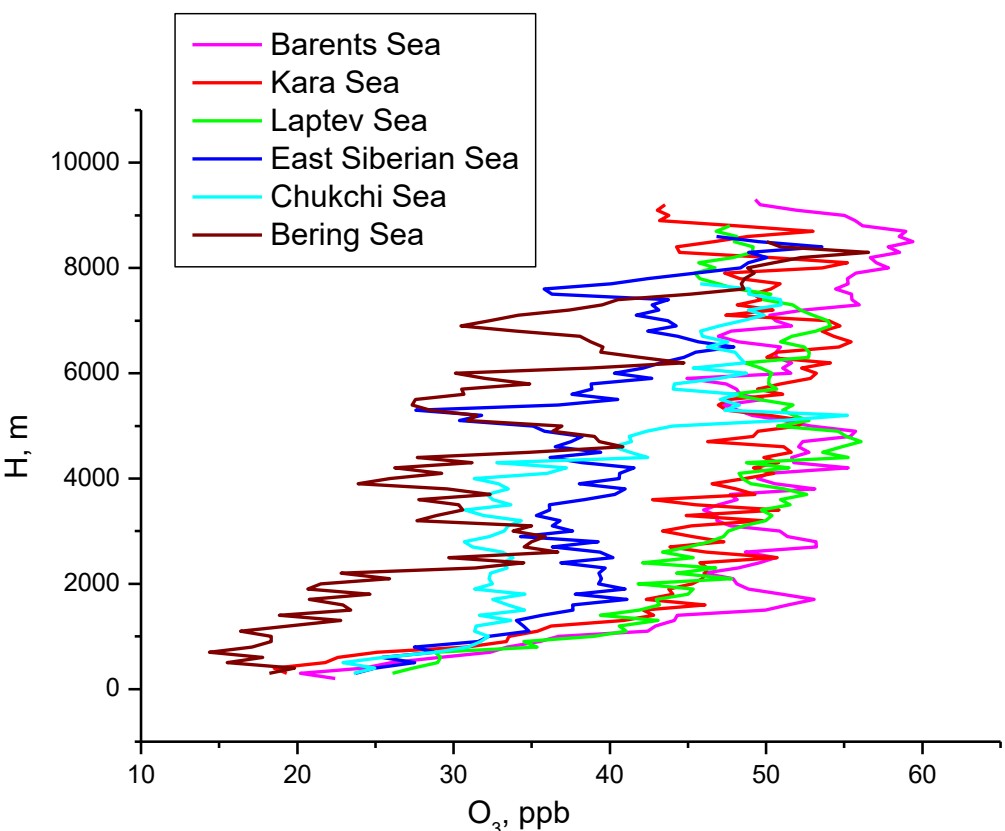

**Figure 14: Vertical distribution of ozone over sea areas.**

It can be seen from Fig. 14 that the ozone concentration in the near-water layer sharply decreases over all seas. Since O₃
belongs to insoluble gases, in contrast to CO2 (Glinka, 1985), this is not the effect of its absorption by the ocean, but the absence of its photochemical formation. In such areas, the main source of O₃ inflow into the troposphere is the stratosphere (Antokhin and Belan, 2013; Berchet et al., 2013). It is possible that the lockdown due to the coronavirus pandemic, as was noted in (Steinbrecht et al., 2021), also affected the situation. At the same time, in the middle troposphere, the eastern and western sectors of the Russian Arctic can be clearly separated. It can be seen that the ozone concentration over the western
sector is much higher. Probably, this is due to the inflow of ozone-forming compounds from Western Europe.

Next, consider the vertical distribution of carbon monoxide. The studies of the CO concentration in background areas in the preindustrial period and now show that it has increased. Thus, in the preindustrial period, it was 90 ppb in Greenland and 55 ppb in Antarctica (Delmas, Legrand, 1998). In (Assonov et al., 2007), a lower value of 38±7 ppb was obtained for Antarctica. At present, the CO concentration averages 140 ppb in the northern hemisphere and 50 ppb in the southern





hemisphere (Karol, 2002).

The data of Fig. 15 demonstrate that the CO concentration during the experiment was close to the background level. CO concentration ranged within 55-118 ppb, which is far lower as compared to continental regions (Davydov et al., 2019; Shtabkin et al.,2016) and close to the values typical for remote regions of Antarctica (Ustinov et al., 2019).

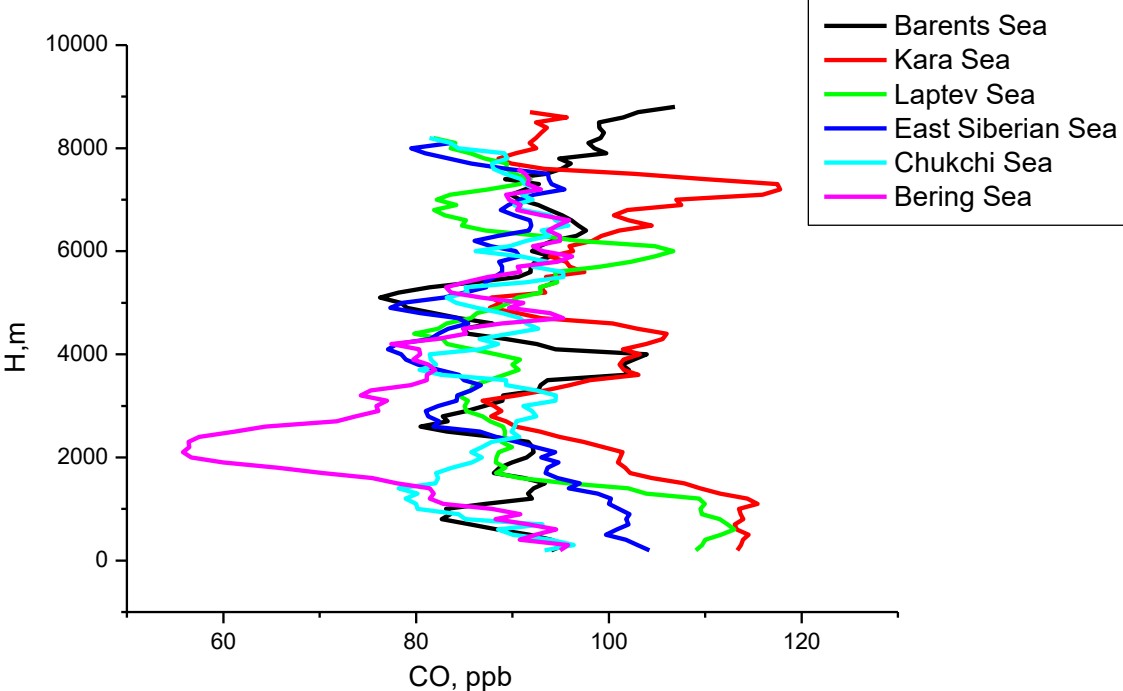

**Figure 15: Vertical distribution of carbon monoxide over sea areas.**

In addition to the above gases, NO and $NO_2$ joined by a single abbreviation $NO_x$ were also measured in the experiment. These gases have both natural and anthropogenic sources (Seinfeld, Pandis, 2006). Recent studies show that efforts of the world community to reduce emissions of these gases have borne fruit. The concentrations of these compounds dropped sharply in urbanized regions, not to mention background ones (Galloway et al., 2014; Lefohn et al., 1999). The data of

Fig. 16 confirm this information.





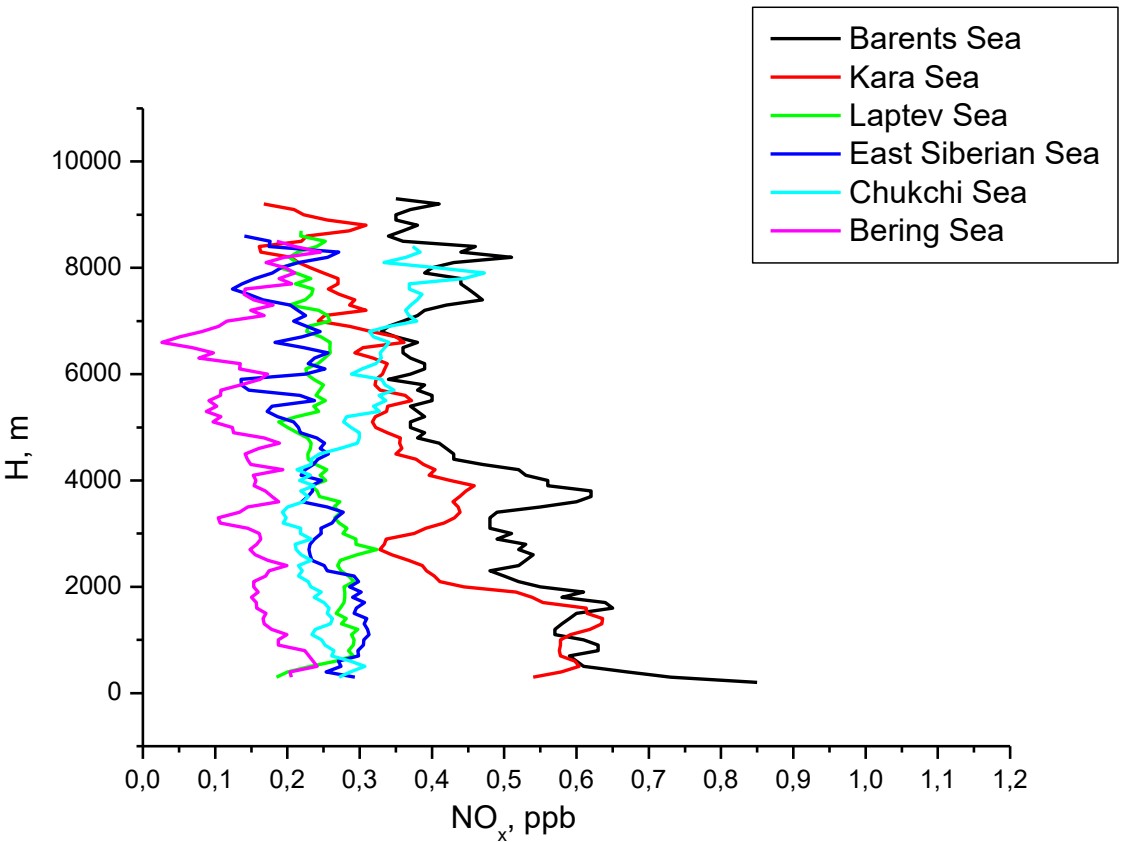

**Figure 16: Vertical distribution of nitrogen oxides over sea areas.**

The $NO_x$ concentrations shown in Fig. 16 range within 0.02-0.90 ppb, which is close to the detection limits (0.01 ppb) of the
devices used.

### 4.2 Aerosol composition

#### 4.2.1 Disperse composition of aerosolReferences

The aerosol instrument suite of the Optik Tu-134 aircraft laboratory allows the aerosol distribution to be measured in a wide
size range with a good resolution (20 bins in the range from 0.003 μm to 0.2 μm and 31 bins in the range from 0.25 μm to 32
μm). To reveal regional features in the vertical distribution in the the troposphere, we analyzed profiles of number
concentrations of the following typically used size ranges: nucleation mode (0.003 μm<Dp<0.025 μm), Aitken mode (0.025
μm<Dp<0.1 μm), accumulation mode (0.1 μm<Dp<1.0 μm), and coarse mode (1.0 μm<Dp<32 μm).





In general, during the campaign, various types of the vertical distribution typical for both coastal marine, polar, and even remote continental regions (Jaenicke, 1993) were recorded. Some concentration profiles, especially for particles of the

nucleation and Aitken modes, had Z-shaped structure described in (Schröder et al., 2002).

Figure 17 shows the profiles of the number density of the main aerosol modes averaged for each water area of the Russian Arctic seas, where flights of the Optik Tu-134 aircraft laboratory were carried out at an altitude from 0.2 to ≈ 9.0 km.

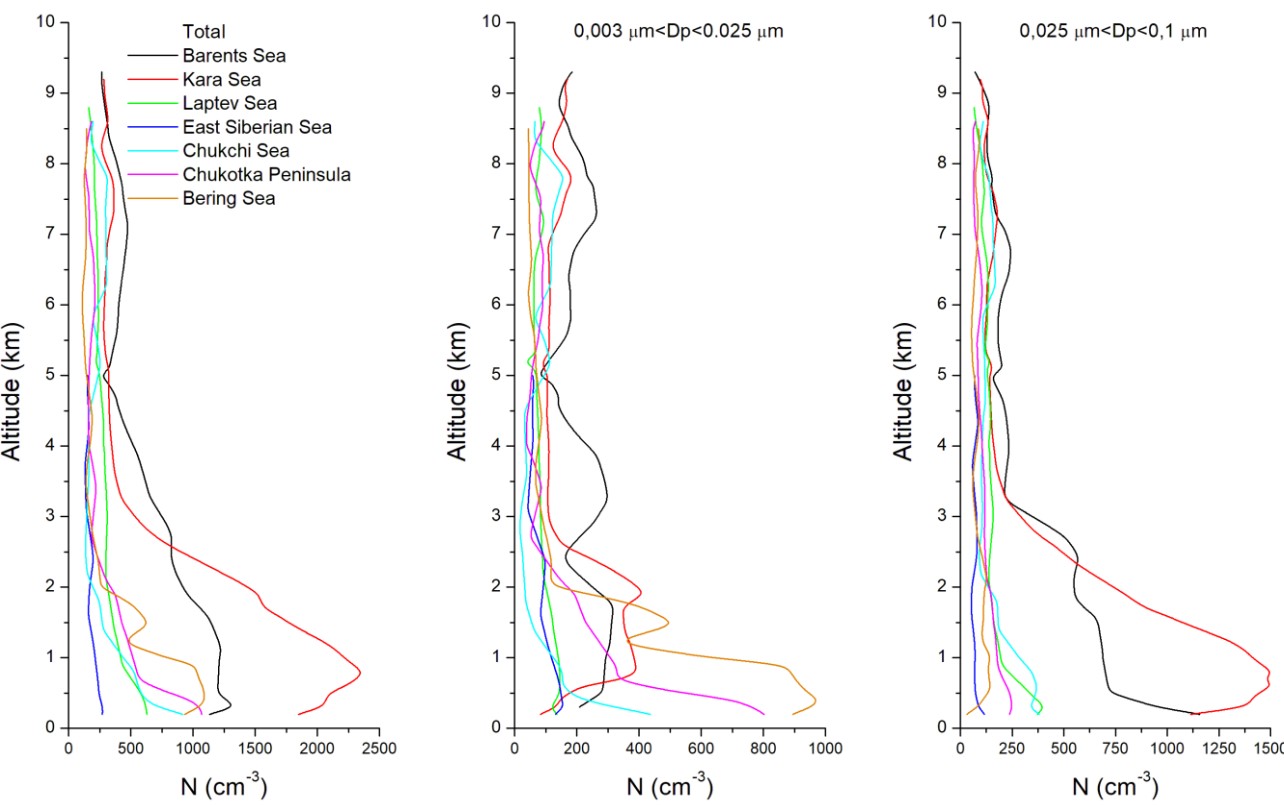



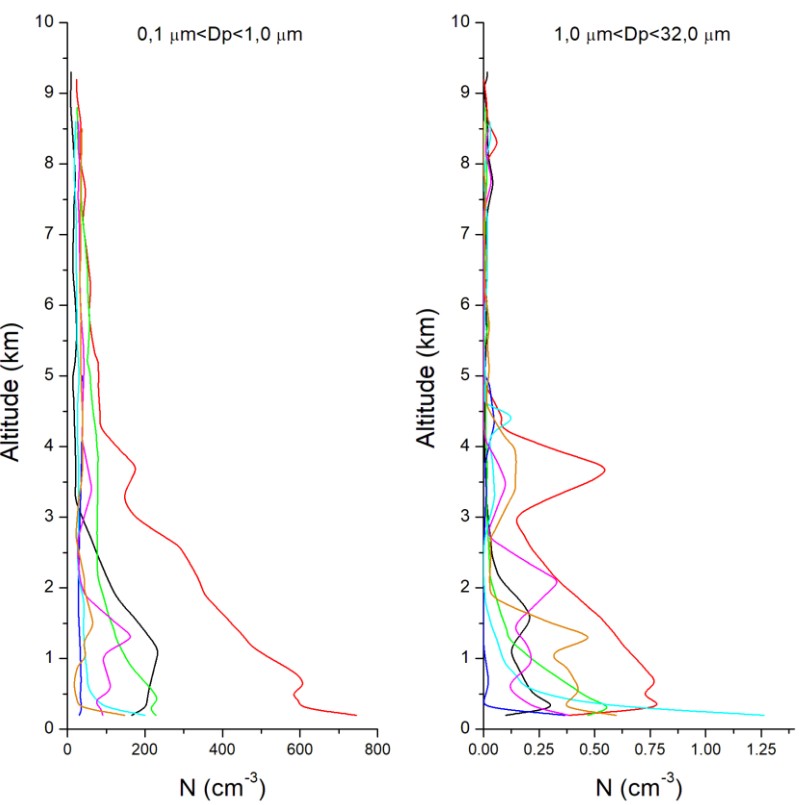

**Figure 17: Average vertical profiles of the total, nucleation, Aitken, accumulation, and coarse mode particle number**

**density over different regions of the Russian Arctic in September 2020.**

It can be seen that in the western part of the Russian Arctic (over the Barents and Kara Seas), the vertical distribution of the continental type was observed. In this distribution, the number density is maximal near the surface, decreases with height to background aloft at altitudes of 2-3 km, and then varies slightly in the free troposphere. At the same time, in horizontal upper flights over these areas, we sometimes recorded number densities comparable to that near the surface. Analysis of the

HYSPLIT back trajectories (Fig. 12) showed that the air masses came to this region from the continent during our flights. That is why the vertical distributions of the continental type were observed.

For the seas of the Asian part of the Arctic, low number densities, occasionally and slightly exceeding the level of 1000 cm$^{-3}$ in the Bering Sea region, were typical. The number concentrations varied with height within a relatively narrow range. It should be noted that during the flight campaign, this territory was affected by a large cyclone with developed cloudiness and

precipitation that led to the washout out of a significant part of aerosol particles from the atmosphere (Fig. 9). Profiles of this type were observed during the ASCOS 2008 summer Arctic campaign conducted by European and American researchers in the Svalbard region (Kupiszewski et al., 2013).

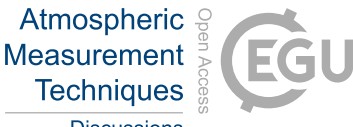

To consider how the size distribution of aerosol particles changes with height, let us turn to Fig. 18, which shows the size spectra averaged in the tropospheric layers with a step of 1000 m, starting from an altitude of 500 m.





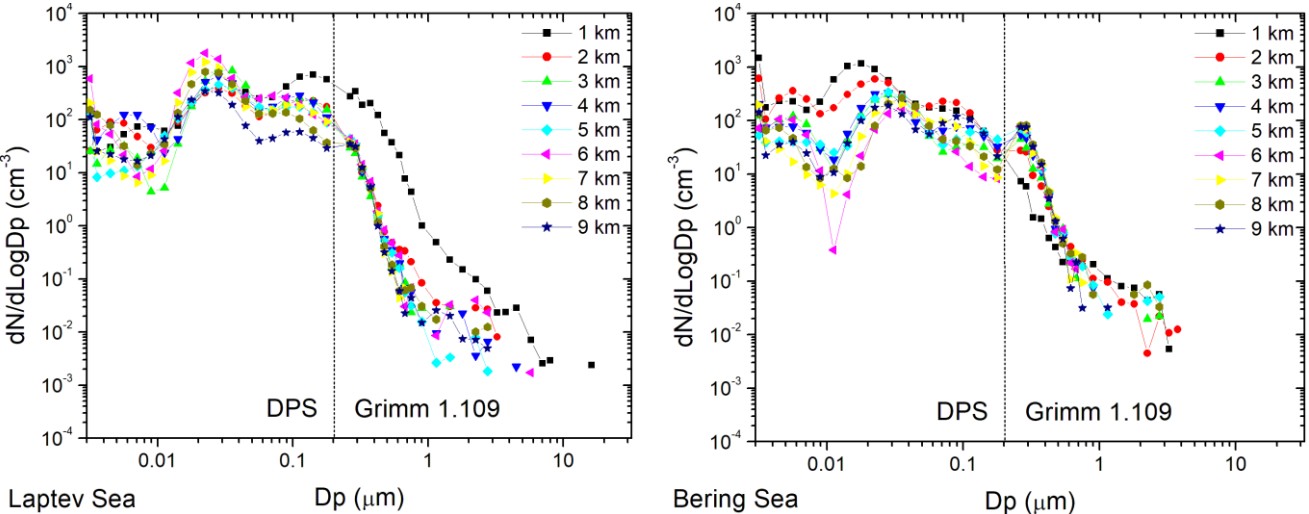

**Figure 18: Average particle size distributions observed at different heights over different regions of the Russian Arctic in September 2020.**

As in the case of vertical profiles of the number density, regional differences are clearly visible here. The size distribution in the western part of the Russian Arctic changes more dynamically with height (Barents and Kara Seas) both in number density and in the intensity and position of the main modes. The particle size spectra were noticeably wider here than in the eastern part of the Russian sector of the Arctic.

In the eastern part, marked changes were observed only in the nucleation size range ($Dp < 0.025$ μm). The weather conditions observed during the flights in this area led to significant washout of aerosols and purification of the atmosphere, which is clearly seen from the complete absence of coarse particles in the size spectra. The intensity of the Aitken mode and the accumulation fraction, which make up the main part of the total number density, did not undergo significant changes with altitude. As a result, the relatively uniform vertical distribution of the aerosol particle number density was formed over the water areas of the Chukchi and East Siberian Seas.

It should be noted that despite the existing gap in detection of particles in the size ranges of 0.2 μm and 0.25 μm by DPS and Grimm spectrometers, the distributions obtained with them match well. The data obtained during our Arctic campaign can be considered unique, because, in the most international aircraft studies, aerosol particles with $Dp < 0.1$ μm are measured with a significantly poorer resolution, and in the Russian sector of the Arctic this study was carried out for the first time.

### 4.2.1 Inorganic aerosol

Every aerosol sample taken in our Arctic experiment consisted of three exposed filters of the AFA-HA-20 type. Two of three filters were analyzed for elemental and ionic composition. In total, 77 samples were taken, of which 52 samples over the seas and the coastal zone of the Arctic.



For the generalized analysis of the chemical composition of atmospheric aerosol relying on the geographical principle and the uniformity of synoptic conditions during the experiment in every area, the entire region was divided into the western (Barents and Kara Seas) and the eastern (Laptev, East Siberian, Chukchi, and Bering Seas) parts. The number of samples for these two parts at different heights is given in Table 5.

Table 5.

Statistics of samples over different regions of the Russian Arctic in September 2020

| Layer height asl | Western Arctic | Eastern Arctic |
|---|---|---|
| 100-300 m (above sea) | 4 | 8 |
| 500-800 m (above coast) | 4 | 1 |
| 4700-5300 m | 6 | 8 |
| 8-10 km | 10 | 11 |

The geometric mean concentrations of elements and ions in the composition of aerosol sampled onto the Petryanov AFA-HA-20 filters are given in Table 6. The concentrations were determined by atomic emission spectroscopy and the HPLC method. The data of Table 6 demonstrate that the synoptic conditions and the origin of air masses affected the aerosol

chemical composition. The continental origin of air masses over the western part of the Russian sector of the Arctic during the experiment manifested itself in a significant increase in the mineral (elemental) component, especially in the main flow, at an altitude of 5000 m. In general, the terrigenous component determines the composition of this layer and dominates in the boundary layer. In the eastern Arctic, the fraction of the ionic component of aerosol increases significantly, although directly in the near-water layer of 100-300 m, its content decreases in comparison with the western part. In the eastern part, the

concentration of mineral forms of alkaline earth metals falls below the detection limit and the concentration of their soluble forms increases significantly. The aerosol acidity in the eastern part at altitudes up to 5000 m inclusive is significantly reduced as compared to the western Arctic.

The total concentrations of ions and elements in the upper troposphere are close for the considered regions. However, for some components, the spread in the concentrations is sometimes very significant. A significant difference in the silicon

content between the regions may indicate that in the upper layers there is a transport of aerosol from the East Asian deserts to the eastern Arctic.






Table 6.

Geometric mean concentrations of chemical components: elements and ions in aerosol (ng/m$^3$)

| H = | western part of Russian Arctic | | | | eastern part of Russian Arctic | | | |
|---|---|---|---|---|---|---|---|---|
| | 200m(sea) | 700m(coast) | 5000m | 9000m | 200m(sea) | 700m* | 5000m | 9000m |
| Si | 973.401 | 1046.028 | 722.022 | 275.071 | 1260.384 | 276.665 | 411.132 | 1364.774 |
| Fe | 627.028 | 321.148 | 252.061 | 248.686 | 605.116 | 361.991 | 207.375 | 73.576 |
| Ca | 107.511 | 151.224 | 784.966 | 324.701 | < | < | < | < |
| Al | 226.173 | 378.672 | 247.806 | 86.613 | 210.636 | 138.332 | 98.310 | 63.291 |
| Cu | 229.707 | 86.904 | 121.404 | 57.051 | 287.561 | 329.670 | 87.154 | 44.524 |
| Ti | 348.624 | 123.944 | 487.214 | 121.992 | < | < | 155.170 | < |
| Mg | 50.171 | 113.378 | 54.398 | 27.122 | < | < | < | < |
| Ni | 14.283 | 26.311 | 6.856 | 9.351 | 21.185 | 38.785 | 3.013 | 10.193 |
| Mn | 21.586 | 19.670 | 10.677 | 5.244 | 21.404 | 22.883 | 11.021 | 9.622 |
| B | 2.010 | 10.172 | 9.923 | 8.067 | 11.973 | 20.621 | 13.026 | 7.924 |
| Cr | 12.150 | 4.132 | 9.107 | 10.439 | 8.731 | < | 16.611 | 8.405 |
| Sr | 1.150 | 47.544 | 3.890 | 5.022 | 4.705 | 1.222 | 1.882 | 2.270 |
| Mo | 8.235 | 1.936 | 9.382 | 2.163 | 6.912 | 6.723 | 10.384 | 7.417 |
| Pb | 5.739 | 2.101 | 9.210 | 2.158 | 13.253 | 5.365 | 10.488 | 1.634 |
| Sb | 5.843 | 3.827 | 15.750 | 0.957 | 0.984 | < | 2.580 | 0.598 |
| Ba | 1.352 | 2.131 | 0.951 | 1.382 | 4.872 | 0.452 | 2.217 | 0.763 |
| Co | 1.736 | 0.068 | 2.335 | 0.384 | 1.807 | 2.450 | 0.346 | 0.684 |
| Sn | 1.054 | 0.692 | 0.687 | 0.647 | 1.214 | 0.549 | 1.090 | 0.581 |
| Zr | 0.560 | 0.884 | 1.265 | 0.207 | 1.556 | < | 0.986 | 0.723 |
| Cd | 0.329 | 0.687 | 0.536 | 1.456 | < | < | 0.817 | 0.131 |
| V | 0.024 | 0.051 | 0.062 | 0.057 | 0.550 | 0.593 | 0.643 | 0.253 |
| Ag | 0.001 | 0.017 | 0.013 | 0.030 | 0.013 | 0.010 | 0.024 | 0.018 |
| Be | 0.003 | 0.010 | 0.012 | 0.002 | 0.011 | < | 0.006 | 0.006 |
| SO$_4^{2-}$ | 320.448 | 479.813 | 77.580 | 78.672 | 142.545 | 340.077 | 65.114 | 40.927 |
| Ca$^{2+}$ | 74.424 | 114.296 | 10.725 | 36.070 | 103.337 | 316.394 | 173.147 | 41.605 |
| NH$_4^+$ | 180.968 | 46.869 | 63.131 | 80.469 | 40.401 | 217.335 | 22.206 | 7.312 |
| Cl$^-$ | 64.491 | 35.205 | 35.248 | 23.999 | 251.375 | < | 95.742 | 44.500 |
| Na$^+$ | 56.686 | 14.719 | 59.913 | 22.010 | 107.584 | < | 74.045 | 22.931 |
| K$^+$ | 37.550 | 61.974 | 17.815 | 9.018 | 39.193 | 30.562 | 30.579 | 11.872 |
| Br$^-$ | 16.602 | 9.199 | 44.996 | 67.057 | 7.585 | < | 76.275 | 3.884 |
| NO$_3^-$ | 6.058 | 4.664 | 27.927 | 4.135 | 4.117 | 143.137 | 15.029 | 0.446 |
| Mg$_2^+$ | 25.639 | 35.744 | 8.780 | 11.270 | 20.103 | 35.664 | 18.710 | 13.817 |
| Li$^+$ | 47.529 | 13.611 | 23.096 | 17.730 | 20.784 | 5.460 | 19.954 | 5.168 |
| NO$_2^-$ | 21.680 | 22.379 | 15.770 | 7.765 | 26.421 | < | 4.240 | 9.570 |
| F$^-$ | 11.286 | 15.982 | 4.628 | 3.786 | 11.515 | 13.178 | 10.412 | 4.386 |
| CH3SO3$^-$ | 1.256 | 4.751 | 0.619 | 4.867 | 6.486 | < | 4.587 | 3.775 |
| H$^+$ | 1.050 | 0.649 | 0.368 | 0.054 | 0.411 | 0.062 | 0.073 | 0.288 |

* concentrations of a single sample over the mouth and delta of the Lena River and the coast of the Tiksi Bay

## 4.2.3 Organic aerosols



A total of 19 samples (six samples in the 0-2 km layer, the others in the 3-9 km layer) were taken during the experiment. A relatively small number of samples is caused by the low concentration of the organic aerosol component. Thus, we had to

pump a large volume of air through the filter to obtain a representative sample of the analyzed material.

As a result, the vertical distribution of the organic component has the form of two layers (Fig. 19).

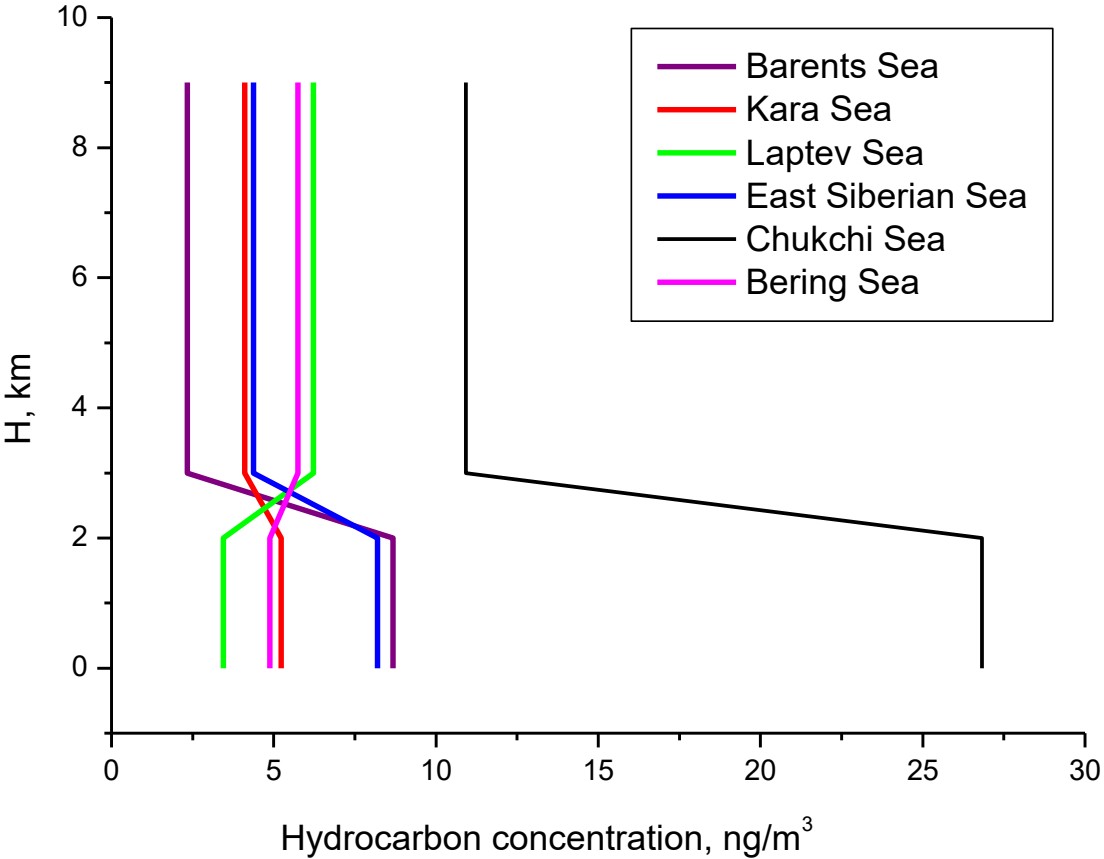

**Figure 19: The total concentration of normal hydrocarbons in the air.**

Figure 19 shows that the greatest amount of organic matter in aerosol was observed over the Chukchi Sea, both in the

boundary layer and in the free troposphere. Over the other seas, the distribution was somewhat different. Thus, over the Barents Sea, the content of organic compounds in aerosol particles in the boundary layer is the second in mass, after the Chukchi Sea, but in the free troposphere it is the smallest of all the seas. Over the Bering Sea, the vertical distribution is even reversed. There is more organic matter in the free troposphere than in the boundary layer. This is probably due to





peculiarities of the transport of organic compounds to this region from adjacent territories. However, a more detailed
analysis is needed for an accurate conclusion.

### 4.2.4 Bioaerosols

From our point of view, total protein is the most adequate marker of the biological origin of organic material in aerosol.
Various proteins are essential materials for the "construction" of biological objects. They are parts of all cells of animals,
plants, representatives of other kingdoms, including all microorganisms. Therefore, their presence in aerosol clearly indicates
the biological origin of this material. The total protein concentration in 22 samples taken over the Arctic seas was determined
by the method described in (You et al., 1997). It should be noted that the total protein concentration in air over the Arctic
seas is low, averaging $10.0 \pm 16.5$ ng/m$^3$. This is significantly lower than the total protein concentrations observed at
different times at different heights in the southwestern Siberia (Kutsenogii, 2006; Agranovski, 2010): $460 \pm 30$ and $1390 \pm$
$1480$ ng/m$^3$, respectively.

Of all the analyzed samples, the sample taken on September 16, 2020, at the Anadyr - East Siberian Sea flight segment at an
altitude of 8500 m, demonstrates a significantly higher total protein concentration of 80 ng/m$^3$ than the concentrations
detected in other samples (no higher than 18 ng/m$^3$. The back trajectory of the air mass carrying this unique sample passed
over Kamchatka four days before sampling and then three days over the Bering Sea at altitudes above 7500 m. Likely, it was
over Kamchatka that the air mass received a relatively large amount of total protein.

Without this point, all other measurement results are shown in Fig. 20.



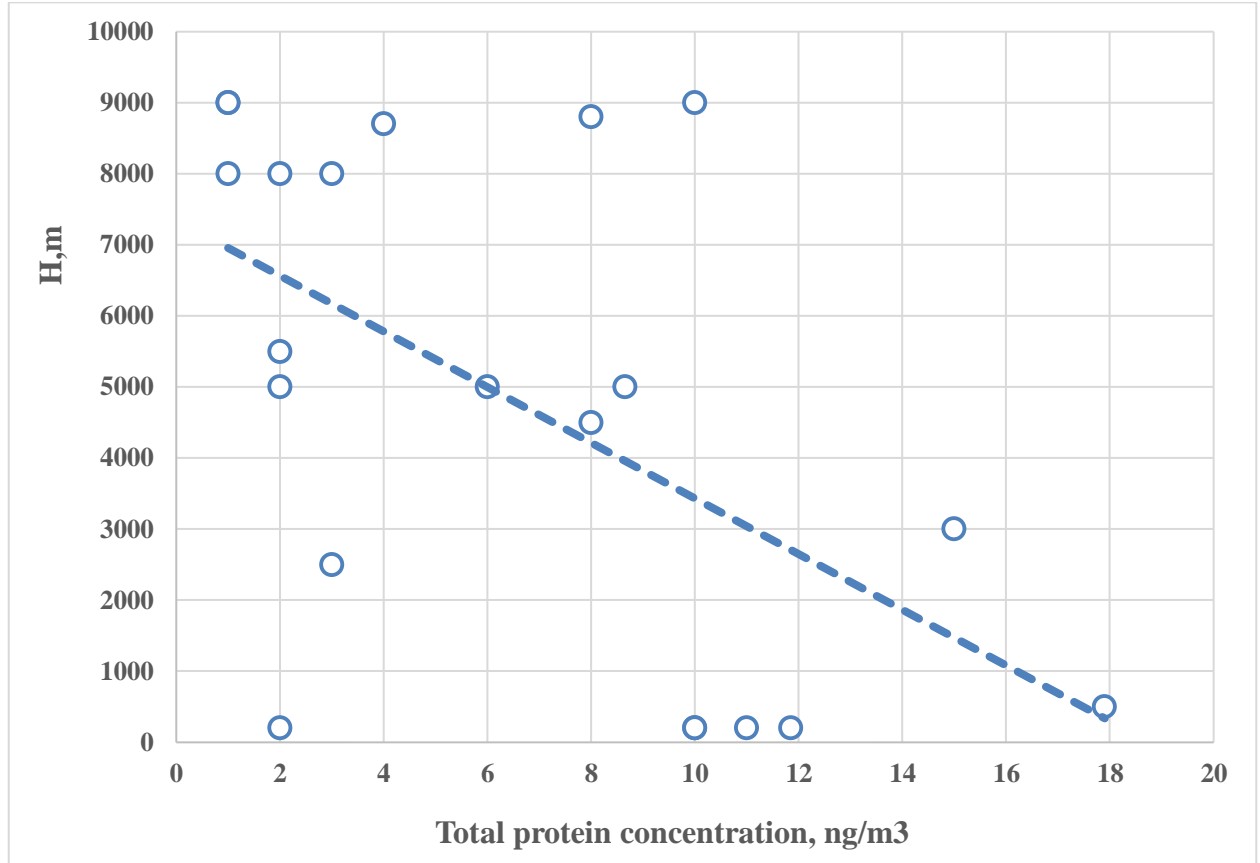

**Figure 20: Total protein concentration in atmospheric aerosol sampled over the Russian Arctic seas.**

As can be seen from the figure, the concentration of total protein in the atmospheric aerosol sampled over the Arctic seas of Russia generally decreases with height. According to the constructed trend, it decreases by more than an order of magnitude as the altitude above sea level increases from 500 to 7000 m. It should be noted, however, that several samples have concentrations below the definition limit of the used method (on the order of 1 ng/m$^3$). The dependence of this concentration on the sampling coordinates has not been revealed.

Let us now turn to the results of studying the microbiological component of atmospheric aerosol in samples taken over the Arctic seas. Figure 21 summarizes all the data on the concentrations of cultured microorganisms in the samples.

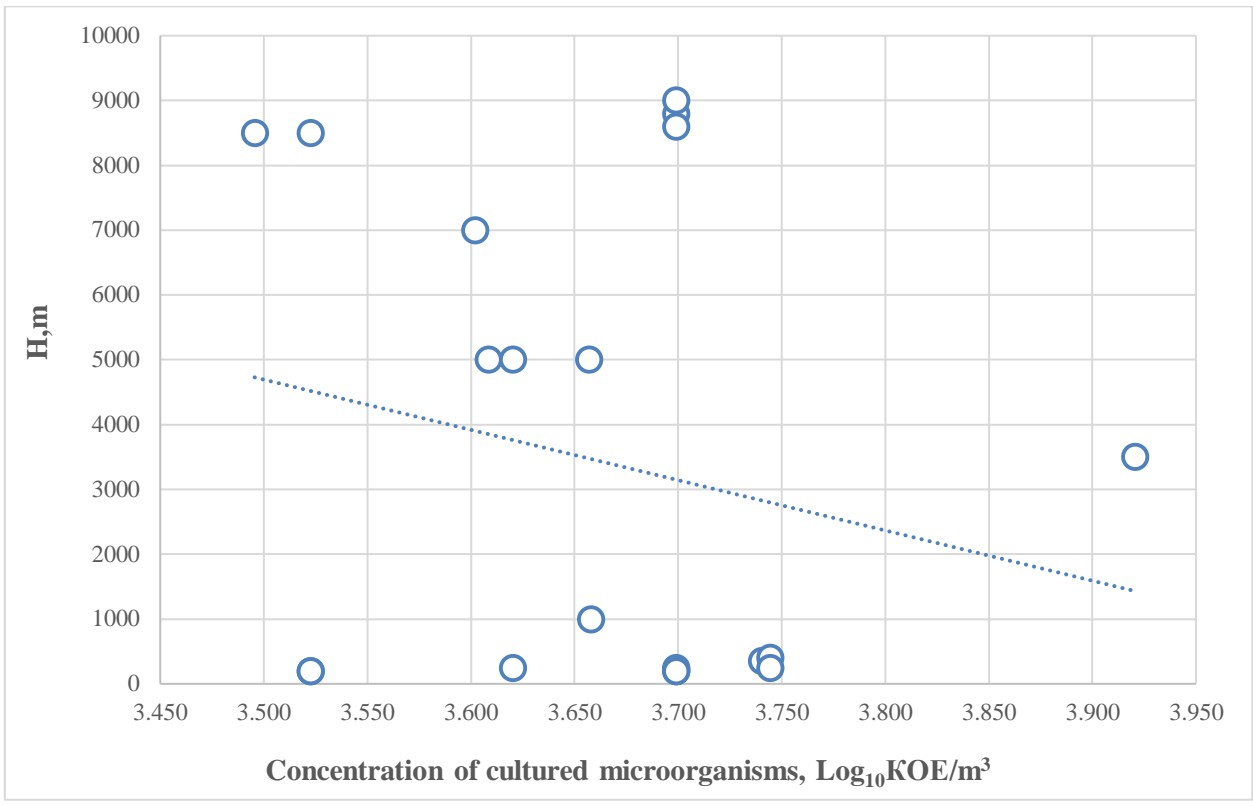

**Figure 21: Concentration of cultivated microorganisms (colony-forming units, CFU) in atmospheric aerosol sampled over the Russian Arctic seas.**

As follows from the figure, the observed concentrations of cultured microorganisms in the atmospheric aerosol sampled over the Arctic seas change insignificantly, from 3000 to 8000 CFU per cubic meter of air. These values are comparable to those previously obtained in Western Siberia (Agranovski, 2010; Andreeva et al., 2019). The vertical profile of the concentration of cultivated microorganisms in the atmospheric aerosol sampled over the Arctic seas of Russia is not significant due to the rather large scatter of the experimental data.

Various bacteria predominated among the identified microorganisms. Fungi in the samples usually accounted for less than 4% of all microorganisms, but one sample contained 38.5% of fungi. Coccal forms of bacteria, less often spore-forming bacteria and non-spore-bearing rod-shaped bacteria prevailed among bacteria in the most samples. It was not possible to reveal the dependence of the representation of various bacterial genera on the sampling height and site.

**4.2.5 Aerosol scattering (aerosol mass concentration)**

In this paper, we omit the detailed analysis of individual profiles of aerosol scattering coefficients over sea areas and in the coastal zone near the aircraft landing points (Fig. 22).





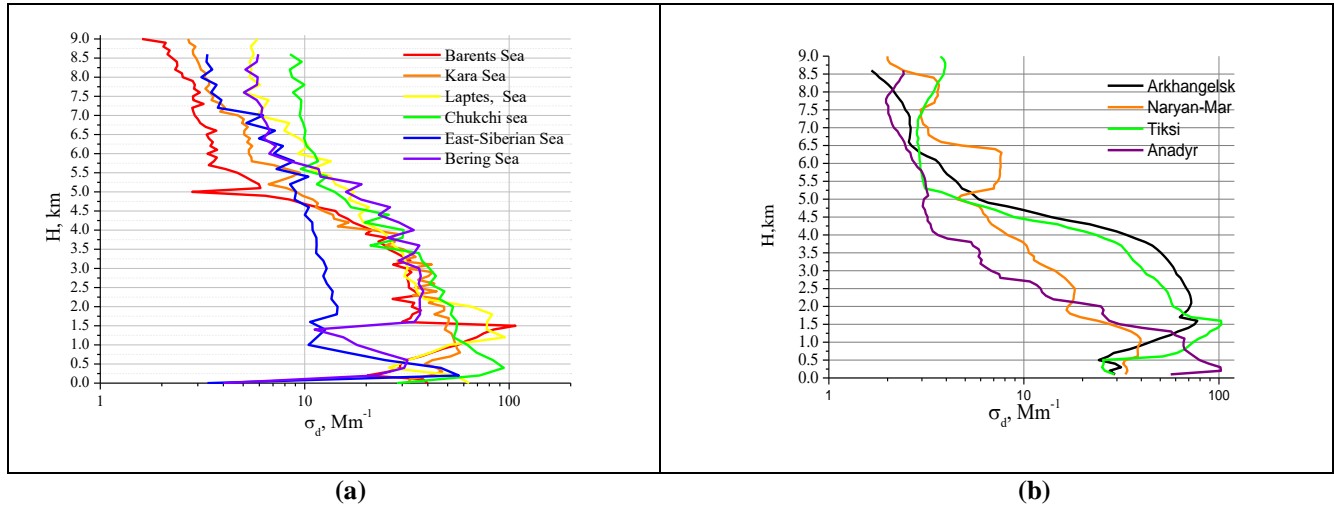

(a)                                            (b)

**Figure 22: Vertical profiles of scattering coefficients on September 04-17, 2020: over (a) sea and (b) coastal areas.**

To be noted is that despite significant variability (by more than an order of magnitude), the characteristic features of the
vertical distribution of the scattering coefficient allow us to assume the continental origin in almost all cases. The concentrations are maximal near the surface. At the same time, in contrast to the classical continental profile, here we observed a blurred boundary of the mixing layer and a significant decrease in $\sigma_d$ ($\lambda = 0.53$ µm) in the altitude range of 0.5-1.5 km. Both of these facts can apparently indicate that the air masses coming to the observation areas in this period and their filling with submicron aerosol were formed over a long time in a rather remote territory. The significant lifetime of
submicron particles in the atmosphere is also evidenced by the drop in $\sigma_d$ ($\lambda = 0.53$ µm) in the lower atmosphere, which was caused by the purification of the lower atmosphere as a result of repeated exposure to cloudy and subcloud washout in the observed weather situation.

**4.2.6 Vertical distribution of Black Carbon**

Figure 23 shows the profiles of concentration of the absorbing substance in the composition of aerosol particles. It is obvious
that the main features of the vertical distribution of the absorbing substance in every realization generally correspond to similar profiles of the scattering coefficients. The maximum values in the surface layer were recorded in the Barents and Kara Seas, as well as over Arkhangelsk and Naryan-Mar. It should be noted that a mixing layer was observed in four realizations recorded over Arkhangelsk, Naryan-Mar, Sabbeta, and the Barents Sea. In other situations, it was practically invisible. This is a quite clear result, since as the air mass moves over the territories having no permanent powerful sources
of black carbon in the surface layer, the formation of the total composition of the submicron aerosol is governed by natural processes. However, the bulk of absorbing particles above the mixing layer enters the region from remote industrial areas (Kozlov et al., 2016). This was most pronounced in measurements in the Chukchi Sea, where the inverted nature of the



vertical BC profile was observed once again. The BC concentration steadily increased from the surface layer to heights of 6-7 km. Similar vertical profiles of absorbing aerosol were recorded in the Chukchi Sea region in our studies within the framework of the POLARCAT campaign in 2008 (Kozlov et al., 2016).

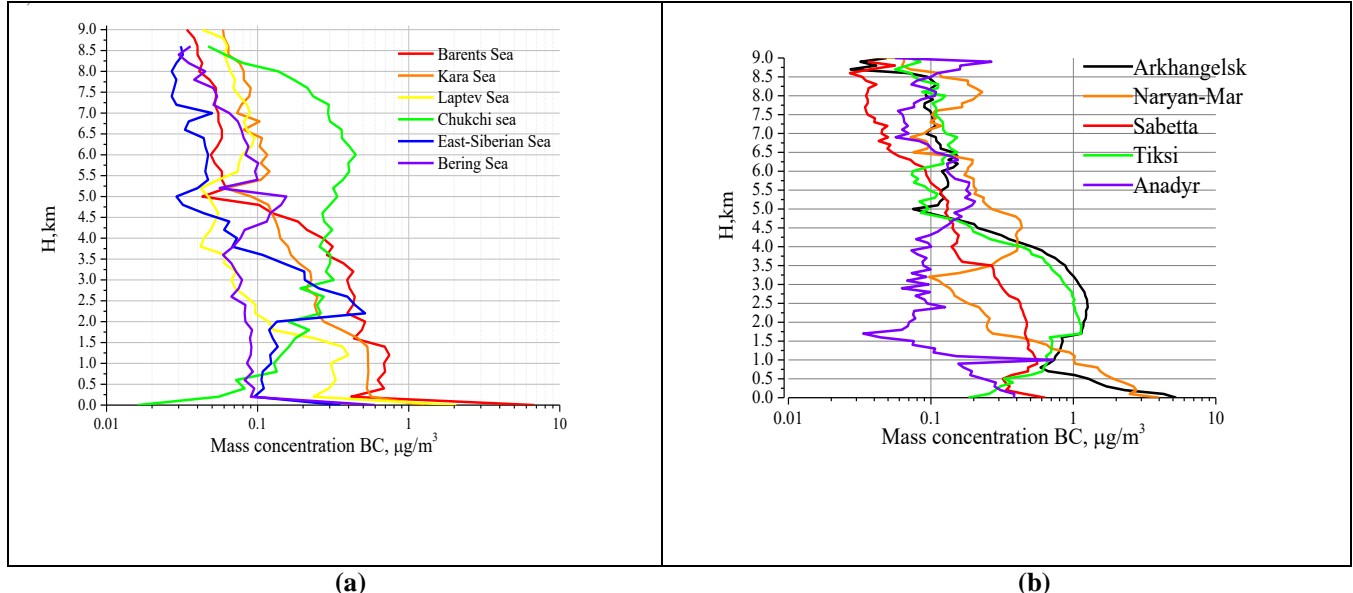

**Figure 23: Vertical profiles of the BC mass concentration on September 04-17, 2020: over (a) sea and (b) coastal areas.**

Comparing the data of measurements of the scattering coefficients and BC concentration, we note that in the most cases, $\sigma_d$ and BC recorded in the lower five-kilometer layer over coastal areas exceed several times the values recorded over the sea.

**4.3 Lidar data on sensing of the water surface**

In flights over the Arctic seas at altitudes of ~ 200 m, the water column was sensed with a LOZA-A2 lidar. The extinction coefficient in water, which is one of the primary characteristics of water optical properties, was retrieved from the recorded lidar profiles of echo signals of the polarized channel at 532 nm. The values of the extinction coefficient allow estimating the degree of water pollution or turbidity. The relative content of chlorophyll can be found from measurements of the signal of laser-induced fluorescence (LIF) of the photosynthesizing pigment of phytoplankton (chlorophyll-a) in the 680-nm channel. This parameter characterizes phytoplankton biomass and is a key characteristic for calculating the productivity of the ocean and seas. In connection with global warming, some scientists predict an increase in the biological productivity of the Arctic Ocean (Babin, 2020; Lewis et al., 2020; Pandolfii et al., 2020). That is why the study of this characteristic in the experiment was very important.

The results of sensing of the surface water column are generalized in Table 7.



Table 7.
Water extinction coefficient in the upper layer of the ocean and the presence of organic matter in the seas of the Russian sector of the Arctic

| Sea | Extinction, m$^{-1}$ | LIF, rel.units |
|---|---|---|
| Barents Sea | 0.104 | 1.00 |
| Kara Sea | 0.134 | 1.59 |
| Laptev Sea | 0.125 | 1.79 |
| East Siberian Sea | 0.128 | 1.42 |
| Chukchi Sea | 0.151 | 1.36 |
| Bering Sea | 0.178 | 1.29 |


It should be noted that the data were obtained on limited sections of the flight route and cannot characterize representatively the average optical properties of water throughout the sea. Nevertheless, they are quite informative for the joint analysis of the spatial distribution of the extinction coefficient in water and chlorophyll fluorescence obtained in simultaneous measurements. The upper water layer in the Chukchi and Bering Seas was the most turbid during the experiment. The

Barents Sea turned out to be the most transparent. The differences in extinction varied more than 1.5 times.

A tendency to an increase in chlorophyll fluorescence in more transparent waters was observed in all the measurements, except for the Barents Sea. Apparently, the turbidity of the ocean was determined by the nonorganic component. The data of sensing of the optical properties of water in the Barents Sea had minimal values. They were used to normalize the relative variation of the phytoplankton concentration. The mostly maximal LIF values were recorded near the coastline, as, for

example, near Vaigach Island in the Kara Sea and Faddeevsky Island in the Laptev Sea. These values decreased with distance from the coast.

An unusual, in this relation, spatial distribution was revealed from the results of retriving the optical properties of water from the data of airborne laser sensing in the Bering Sea. Figure 24 shows the plots of the LIF values and the extinction coefficient in the surface water layer at the flight section at an altitude of 200 m in Anadyr Bay on September 16, 2020. It

can be seen from the figure that the extinction coefficient decreases and the LIF signal increases with the distance from the coast. This feature of the spatial distribution of chlorophyll concentration is characteristic of semi-enclosed water areas of bays on the shelf zone of the seas affected by dynamic hydrophysical processes. As was shown in (Navrotsky et al., 2019) based on the analysis of satellite observations of chlorophyll concentration in Peter the Great Bay (Sea of Japan), this spatial distribution "is determined by the effect of currents generating shelf waves, which cause the outflow of phytoplankton

concentration from the coastal zone."





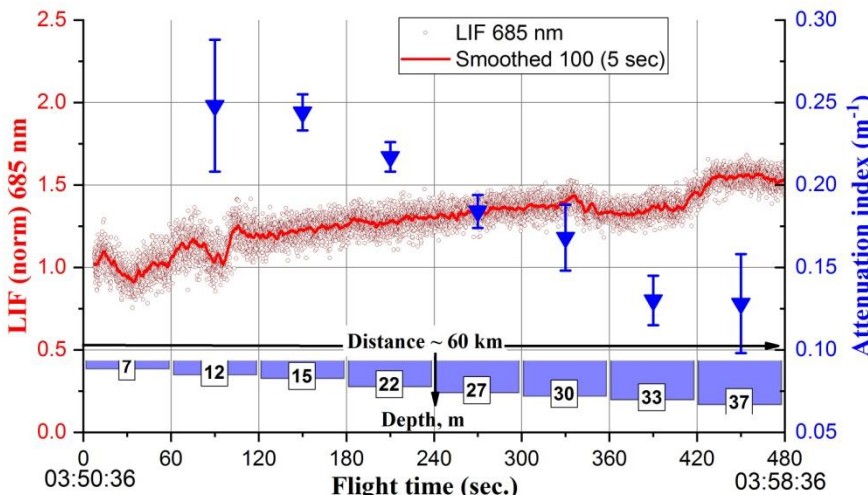

**Figure 24: Relative chlorophyll concentration and the water extinction coefficient in the Anadyr Bay of the Bering Sea according to laser sensing data on September 16, 2020.**

The results shown in Fig. 24 demonstrate the high sensitivity and spatial resolution in determining the optical properties of

the surface water column by airborne laser sensing of the seas.

**5 Conclusions**

This paper presents the tentative results obtained during the special experiment on airborne sensing of the troposphere in the

Russian sector of the Arctic. For this experiment, the equipment of the Optik Tu-134 aircraft laboratory was significantly

supplemented, namely, a NOx gas analyzer, a BC meter (AE-33 aethalometer), and a PSR-1100F spectroradiometer were

installed. The navigation system was replaced with a specially developed CompaNav-5.2 navigation system, which

combined navigation sensors with repeaters. To expand the lidar capabilities, a fluorescent channel was created to record the

laser-induced fluorescence of photosynthetic phytoplankton pigments during the sensing of aquatic environments.

With the modernized measurement system of the Optik Tu-134 aircraft laboratory, the sensing of the air composition was

carried out for a short period (September 4-17, 2020) over all seas and coastal areas of the Russian sector of the Arctic and

the Bering Sea. The concentrations of CO, $CO_2$, $CH_4$, NO, $NO_2$, $SO_2$, $O_3$, aerosol and black carbon have been measured. Air

samples were taken to determine organic and inorganic aerosol components and bioaerosol. The LOZA-2 lidar was used for

sensing of the turbidity of the upper water layer and determination of the phytoplankton concentration in water. Installation

of the spectroradiometer onboard the aircraft laboratory made it possible to determine the spectral characteristics of the water and underlying coastal surface. The data are currently being processed.

The primary results showed that the concentrations of CO, NO, $NO_2$, $SO_2$, $O_3$, aerosol and BC during the experiment had very low values typical for background regions. Low concentrations of $CO_2$ were observed over the seas in the near-surface layer, which indicates the absorption of this gas by the ocean. The methane concentrations recorded over all seas of the Russian sector of the Arctic and the Bering Sea were increased as compared to the coastal areas. Whether the methane was of oceanic origin or was transported from land is not clear yet, since the back trajectories start on the continent. To clarify

this issue, numerical simulation is now being carried out, and the special land-ocean experiment is planned for 2022.

**Author contributions**

APN, AMYu, BBD, BSB, CDG, DDK, FAV, IGA, KAS, KAV, NSV, OOV, SDE, SDV, TGN, ZPN deal with measurements, primary processing and interpretation of data; AG, AIS, AVG, BYuS, KSN, LK, NP, OSE, PMV, PJD, PIE., PIV, RTM, RIK, ROA, SAS, STK were responsible for processing of samples and additional information, interpretation of

results and preparation of some sections of this paper; BVE, MAV, MIA take part in planning and performing flight experiments.

**Competing interests**

The contact authors have declared that neither they nor their co-authors have any competing interests.

**Acknowledgements**

We would like to acknowledge our colleagues from the following organizations for their assistance in organizing and conducting this campaign, and in particular, Laboratoire des sciences du climat et de l'environnement and Laboratoire atmosphères, milieux, observations spatiales (France); Finnish Meteorological Institute and Institute for Atmospheric and Earth System Research, University of Helsinki (Finland); Center for Global Environmental Research at the National Institute for Environmental Studies (Japan); the National Oceanic and Atmospheric Administration, US Department of Commerce

(USA); Max-Planck-Institute for Biochemistry (Germany); and University of Reading (UK). We acknowledge the support of the YAK-AEROSIB programme and the ERA-PLANET iCUPE project.



## Financial support

The project is funded by the Ministry of Education and Science of the Russian Federation, Agreement No. 075-15-2021-934.

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
