# Peer review of "Integrated airborne investigation of the air composition over the Russian Sector of the Arctic"

_Atmospheric Measurement Techniques, 2021_

## Author Comment (AC1)

Response to Reviewer 1

SUMMARY

Belan and co-authors present gas and aerosol measurements, performed from an airborne platform in the Russian sector of the Arctic. As the authors stated, this sort of measurement in the Russian Arctic are extremely rare and, as a consequence, very valuable. Hence, will provide the needed reference for evaluating the performances of global models in a relatively unknown region of the Arctic. Despite the scientific interest and need for these measurements, the manuscript is very confusing, in its language, structure and objectives.

My main concern is related to the choice of the journal and the objective of the manuscript. The measurements are indeed a novelty, not from the technical point of view, but rather for the location. Most of the deployed instruments are commercially available or already described and validated in previous papers. The description of the atmospheric results is very chaotic and no clear conclusions can be drawn. If the manuscript aims to provide a general overview of the experiment, I suggest reducing the discussion to the grand average of each variable measured during the campaign and contextualizing the vertical variability as a function of the synoptic situation and air transport. I can easily see a resulting overview paper being submitted to ACP. The detailed description of each variable during each flight should be done in separate articles with a narrower scientific question. To promote the use of this dataset for modelling purposes, a technical description of the measurements could be, eventually, provided into a separate descriptive manuscript potentially submitted to Earth System Science Data. If I am correct, data must be, then, publicly available.

Considering the importance of the dataset, I hope that the authors will follow my suggestions and reconsider their strategy for publication. In its current status, the manuscript is not suitable for publication on AMT.

Response:

The authors agree with most of reviewer's comments. Indeed, the manuscript is at the intersection of three fields: technical, physico-chemical and environmental. The aim of publication is to provide data about the unique experiment and to characterize generally the air composition in the region not earlier covered by observations. The description of experiment necessarily requires the description of experimental equipment. However, the paper on the basic equipment of the Tu-134 Optik aircraft laboratory has been already published in the Russian journal. At the same time, the equipment was significantly updated during the preparation to the experiment. That is why we decided not to describe the equipment of the aircraft laboratory in a separate paper (since publication partially covering this topic already exists). On the other hand, we cannot provide the measured data without describing the device, they were obtained with. Therefore, we chose a compromise way. We provide general information about the aircraft laboratory with more detailed description of Russian devices, which are not considered in the international literature yet. This manuscript mostly deals with technical or methodological issues and, in our opinion, is best suited for AMT. In the future, we plan to prepare a publication with complete analysis for ACP, as recommended by the reviewer. In the longer term, after checking the data, a free accessible database will be compiled.

SPECIFIC COMMENTS

L45-50: The paper is already long enough. This part is very generic and not needed.

Response: Removed.

L51-55: "Spontaneous question", remove all the references, there are more words in the reference list than in the actual statement. Not of smooth reading.

Response: References are partially removed.

L55-61: This part is unintelligible. Rephrase

Response: The text is rephrased and shortened.

L62-64: I would always suggest avoiding the use of a long list of references. Like it is, this sort of listing does not help the reader to identify a specific citation with a specific result, becoming, as a consequence, not useful. Provide one to three reference for each statement or scientific information, not more. This issue recurs along the entire manuscript.

Response: The number of references here and hereafter in the text is reduced.

L72-79: It is written "there are no systematic observations of the vertical distribution of gas and aerosol components of the atmosphere." In the following lines the vertical measurements of gas and aerosol are described. The writing is not coherent.

Response: The paragraph is removed.

L88: replace "appeared" with "established".

Response: Replaced

L118-122: insert bullet points

Response: The recommendation is unclear.

T1: adjust legend. Range in ppm?

Response: The ranges and uncertainties are given in Table 1 in ppm.

L181: black carbon not in capital letters

Response: Corrected

L182: I think you should use the equivalent black carbon (eBC) following Petzold et al. 2013.

Response: Agree. The correction has been made: black carbon -> equivalent black carbon, BC -> eBC, $M_{BC}$ -> $M_{eBC}$

L183: what is "IAO SB RAS"?

Ответ: V.E. Zuev Institute of Atmospheric Optics. Added to the text.

F4 is not needed, and difficult to interpret

Response: Fig. 4 is removed

L181-213: usually filter based transmission photometers calculate eBC mass from absorption or attenuation coefficient using the mass transmission or absorption cross section. Besides the fact that the authors do not specify which MAC or MTC values they use, they also state "The number concentration of BC particles in the air is calculated by the software". I am curious to know how the number concentration was calculated.

Response: Corrected. The way of calculating is provided.

L215-221: are scattering coefficient corrected for truncation error?

Response: No, the scattering coefficients are not corrected. The nephelometer recorded the values of the angular aerosol scattering starting from the level of molecular scattering about $0.001 \text{ km}^{-1} \text{ sr}^{-1}$.

L228: "This method is traditional and has been described many times in the literature". Provide reference.

Response: The reference is provided: Peregud E.A. , Gorelik D.O. Instrumental methods of air pollution control. Leningrad: Chemistry, 1981. 384 p.

F5 not needed, hard to read. See comment on Figure 4

Response: Figure 5 is removed.

F7 not needed. Does not provide useful information for scientific scope.

Response: Figure 7 is removed.

S2.4-2.5: These two sections could be merged. S2.5 does not provide enough information on the specific sensors, while I genuinely do not understand the description of CompaNav-5.2 in S2.4.

Response: This suggestion is not logical. Section 2.4 describes the spectroradiometer, while Section 2.5 describes the navigation system. The purposes and operating principles of these devices are completely different.

S3.1: irrelevant to the understanding of the manuscript. A short statement on planning change could be introduced in a different section, but it does not need a dedicated chapter.

Response: Section 3.1 is removed.

L355: typically

Response: Corrected

L357: "The minimum height was 200 m above the sea and 500 m above land." Repetition.

Response: Removed.

S3.3: It is important to describe the synoptic conditions of the flight. However, this day-by-day report is unnecessarily long and tedious to read, and could be easily summarized by a table. The authors could then simply describe the difference between the various influence periods.

Response: The text is revised and shortened.

F9: revise this figure, make it easier to read, include flight pattern or interested region.

Response: The number of the maps is reduced.

L425: "This is understandable, since the concentration of carbon dioxide in the atmosphere is increasing all over the world." I would expect some better wording and conclusion.

Response: The text is revised

F10: adjust margin and size of the panels

Response: This seems inappropriate, since the dynamic range decreases.

L422: "However, this is due to the transfer from the continent. To check this, initially unplanned sensing was carried out over the Bering Sea. It confirmed this conclusion." The authors must provide evidence of what is stated. The reader could not verify this information, since is not shown.

Response: The explanation with back trajectories is provided.

F11: provide real legend. It might be one line of text!

Response: Provided

F12: give more info on the backtrajectories.

Response: More info is given.

L470: "This conclusion is, in principle, clear from the above synoptic maps (Fig. 9) and follows from the constructed back trajectories (Fig. 12)." Back trajectories are not discussed. So, I am not sure, what it is clear.

Response: The comparison of the concentrations at the back trajectories starting from the sea and the continent is provided.

F13: to be removed. Not usefull.

Response: The figure is replaced with the better one.

L479-482: I do not see the reasons for mentioning that there is a source of methane that is not detected.

Response: The paragraph is removed.

---

## Author Comment (AC2)

Response to Reviewer 2

Review of Integrated airborne investigation of the air composition over the Russian Sector of the Arctic by Boris Belan et al.

The paper describes measurements of traces gases, aerosol properties and ocean extinction coefficient from an aircraft campaign in the Russian Arctic in Sep. 2020.

The research plane was well equipped and the presented data is valuable and well suited for publication.

The aim of this work is to introduce the measurement campaign and the data. Hence, the paper is a little weaker on the interpretation of the findings. However, given the wealth of data (and the length of the article) I am completely fine with this.

I only found a few minor comments, listed below. Generally the paper is well-written and clearly structured.

Introduction
Lines 51/52: you may give the keyword "Arctic amplification"

Response: The keyword is added.

Instrumentation:
This is a well equipped aircraft for relevant measurements. Could you describe in 1 or 2 sentences what is meant by "with good resolution" in line 170? (e.g. sampling time, insecurities …)

Response: This sentence concerns the ranges of particles that are recorded by the counter. It can be omitted as the ranges are already listed in lines 164 and 165.

Line 265: please write "laser induced fluorescence" to introduce "LIF" at first use

Response: Corrected.

Fig 10: does Karskoye mean Kara Sea? Could you clarify?

Response: Corrected.

Discussion on origin of CH4 line 465 ff: Did you consider that the CH4 may have originated from the ocean? See e.g. here: https://phys.org/news/2021-03-east-siberian-arctic-ocean-elevated.html
If you had seen higher CH4 over lakes in tundra compared to the Arctic Ocean I would be convinced. Fig 12 (hysplit) is good from a methodologic point of view; however, as the sources of CH4 are the ground (land or sea) I am not sure here, whether your reasoning is complete. – Do you have any idea how in (e.g.) Sabetta region the gradient of CH4 in boundary layer looks like when flying from the tundra towards the ocean?

Response: We have the measured gradients between the continent and the Kara Sea and between the continent and the Laptev Sea in the lower 200-m layer. During the experiment, the gradients were directed from the continent to the ocean. These data will be presented in detail in the next paper.

Line 599: I do not understand the sentence: A relatively small number of samples is caused (or impaired?) …

Response: This means that the concentration of organic matter in the atmosphere is low. Therefore, to collect the amount of substance needed for analysis, it is necessary to pump a lot of air. It takes a lot of flight time. Therefore, the number of samples for organic matter is small. And it is small relative to the number of samples for inorganic matter.

Fig 19: can we understand the high values over Chukchi Sea?

Response: We have a publication on this topic. In it, we explain these high values by the transport from Alaska based on the back trajectory method. When this paper was being prepared, this result was not available yet.

Caption of Fig 20 I would repeat in the figure caption that one high data point has been omitted.

Response: Corrected

Line 665: good correlation between scattering and BC. I see this for Arkhangelsk and Tiksi and this is indeed remarkable. But would we expect generally a good correlation between those quantities? I am not sure on this.

Response: Reviewer's remark is absolutely correct. Of course, we cannot expect that these two characteristics in all arrays are well correlated. It is only noted here that the main features of the vertical BC distribution in every realization generally correspond to those of the scattering coefficient, since in these measurements we deal with air masses from remote areas, in which the vertical distribution is formed by general factors during the air mass transport (ageing) over the territory having no powerful sources of particles of various origin.

---

## Author Response (AR2)

Response to Associate Editor

L118-122: insert bullet points, the reviewer means add a bullet point like below, which makes the text better to be understood

☐ $CO_2$, $CH_4$, and $H_2O$ - G2301-m operating based on the technology of cavity ring-down spectroscopy (CRDS, Picarro Inc.,

☐ USA);

☐ $O_3$ – Model 49C UV photometric gas analyzer (Thermo Environmental Instruments 120 Inc., USA);

☐ CO – Model 48C non-dispersive infrared (NDIR) correlation gas-filter analyzer (Thermo Electron Corp., USA);

☐ NO and $NO_2$ (NOX) – Model 42i-TL chemiluminescence gas analyzer (Thermo Fisher Scientific Inc, USA).

Response:  Markers inserted. In the corrected version of line 103-108

L181: black carbon not in capital letters

Please check, it seems that they are still in capital letters in title of section 2.2.2 Equipment for measurement of Black Carbon and aerosol scattering

Response:  Fixed: In the corrected version of line 168.

F9: revise this figure, make it easier to read, include flight pattern or interested region.

The reviewer suggested an update of the figure (not referred to the reduction of total number of the figures in the paper), please make update. The numbers in current figure are not readable (font too small), and the important patters are not highlighted.

Response: Fig. 9, in the corrected version, Figure 6 represents fragments of standard synoptic maps. We cannot increase the size of the numbers on it. We propose to reduce the number of cards and increase the drawing itself.

F10: adjust margin and size of the panels

The reviewer suggested an update of the figure, by reducing the margin of each figure and increasing the font size in each figure, thus better for reading, please update accordingly

Response: Fig. 10, in the corrected version of Figure 7, corrected.

Also corrected Fig. 11 (Fig. 8) as they are similar.